## OPEN

# Visualizing maturation factor extraction from the nascent ribosome by the AAA-ATPase Drg1

Michael Prattes [1,2,10], Irina Grishkovskaya [3,10], Victor-Valentin Hodirnau [4], Christina Hetzmannseder[1], Gertrude Zisser[1], Carolin Sailer[5], Vasileios Kargas[6,7,8], Mathias Loibl[1], Magdalena Gerhalter[1], Lisa Kofler [1], Alan J. Warren [6,7,8], Florian Stengel [5], David Haselbach [3✉] and Helmut Bergler [1,2,9✉]

The AAA-ATPase Drg1 is a key factor in eukaryotic ribosome biogenesis that initiates cytoplasmic maturation of the large ribosomal subunit. Drg1 releases the shuttling maturation factor Rlp24 from pre-60S particles shortly after nuclear export, a strict requirement for downstream maturation. The molecular mechanism of release remained elusive. Here, we report a series of cryo-EM structures that captured the extraction of Rlp24 from pre-60S particles by *Saccharomyces cerevisiae* Drg1. These structures reveal that Arx1 and the eukaryote-specific rRNA expansion segment ES27 form a joint docking platform that positions Drg1 for efficient extraction of Rlp24 from the pre-ribosome. The tips of the Drg1 N domains thereby guide the Rlp24 C terminus into the central pore of the Drg1 hexamer, enabling extraction by a hand-over-hand translocation mechanism. Our results uncover substrate recognition and processing by Drg1 step by step and provide a comprehensive mechanistic picture of the conserved modus operandi of AAA-ATPases.

The assembly of ribosomes is a major activity in all cells. It is evolutionary conserved and tightly coordinated with cell cycle progression and proliferation. Eukaryotic ribosome biogenesis starts in the nucleolus by transcription of a precursor RNA (pre-rRNA) that contains the information for the 18S, 5.8S and 25S ribosomal RNAs (rRNAs) in yeast. Co-transcriptional pre-rRNA cleavage in the nucleolus separates the maturation paths of the small 40S and large 60S ribosomal subunits (reviewed in refs. [1–4]). The pre-60S particles are subjected to massive structural rearrangements during maturation, including removal of pre-rRNA spacers, formation of the polypeptide exit tunnel (PET) and incorporation of the 5S pre-rRNA[5–10]. Thereafter, loading of Nmd3 provides the nuclear export signal (NES) for the exportin Crm1, also known as Xpo1, which mediates transport through the nuclear pore complex[11].

The initial event of cytosolic maturation is the release of the shuttling protein ribosomal-like protein 24 (Rlp24) by the essential hexameric type-II AAA-ATPase diazaborine resistance gene 1 (Drg1)[12–14]. This is a prerequisite for downstream maturation, including release of other shuttling proteins and loading of late joining maturation factors and ribosomal proteins[12–16]. Drg1 contains an amino domain and two nucleotide-binding domains (D1 and D2) per monomer and is closely related to human p97 and yeast Cdc48 (refs. [17–20]). Mutations in SPATA5, the human ortholog of Drg1, cause developmental and neurological defects, underlining the protein's importance for higher eukaryotes[21–24].

The interaction with the unstructured Rlp24 C-terminal domain stimulates ATP hydrolysis in both AAA domains of Drg1, which drives the release reaction[12]. However, only ATP hydrolysis in D2 is essential for viability and release of Rlp24 (ref. [16]).

The D2-domain-specific Drg1 inhibitor diazaborine therefore blocks Rlp24 extraction and prevents cell growth and proliferation[16,17]. The mechanism of Rlp24 release by Drg1 was unknown.

Here, we visualize by cryo-electron microscopy (cryo-EM) how Drg1 is recruited to the pre-ribosome to extract Rlp24. Rlp24 is captured by the Drg1 N domains and AAA domains and extracted by hand-over-hand translocation. Our data uncover orchestrated conformational changes of the AAA-ATPase during substrate translocation and provide a structural basis for the substrate-processing mechanism.

## Results

**Structure of export-competent pre-60S particles.** To unravel the substrate-release mechanism, we determined the structure of Drg1 captured during Rlp24 extraction from the pre-ribosome. We assembled pre-60S particles, purified using tandem affinity purification (TAP)-tagged Bud20 as bait from a leptomycin B (LmB)-sensitive *S. cerevisiae* strain, with separately purified Drg1 in vitro and collected single-particle cryo-EM data (Fig. 1, Extended Data Fig. 1 and Table 1). To gain a more homogenous pre-ribosome population, we enriched the state immediately before nuclear export using the export inhibitor LmB (Fig. 1a). LmB covalently modifies exportin Crm1/Xpo1, which prevents binding to the nuclear export signal (NES) of Nmd3 on the pre-ribosome[25,26]. Therefore, the particles cannot be transported through the nuclear pores and accumulate prior to export (Fig. 1b). These particles closely resemble the natural substrate of Drg1, as confirmed by their similar composition after diazaborine treatment (block after export) (Fig. 1a). The increase in Nmd3

[1]Institute of Molecular Biosciences, University of Graz, Graz, Austria. [2]BioTechMed-Graz, Graz, Austria. [3]Research Institute of Molecular Pathology (IMP), Vienna BioCenter, Vienna, Austria. [4]Institute of Science and Technology Austria, Klosterneuburg, Austria. [5]Department of Biology, University of Konstanz, Konstanz, Germany. [6]Cambridge Institute for Medical Research, Cambridge Biomedical Campus, Cambridge, UK. [7]Wellcome Trust-Medical Research Council Stem Cell Institute, Jeffrey Cheah Biomedical Centre, Cambridge Biomedical Campus, Cambridge, UK. [8]Department of Haematology, University of Cambridge School of Clinical Medicine, Jeffrey Cheah Biomedical Centre, Cambridge Biomedical Campus, Cambridge, UK. [9]Field of Excellence BioHealth - University of Graz, Graz, Austria. [10]These authors contributed equally: Michael Prattes, Irina Grishkovskaya. ✉e-mail: david.haselbach@imp.ac.at; helmut.bergler@uni-graz.at

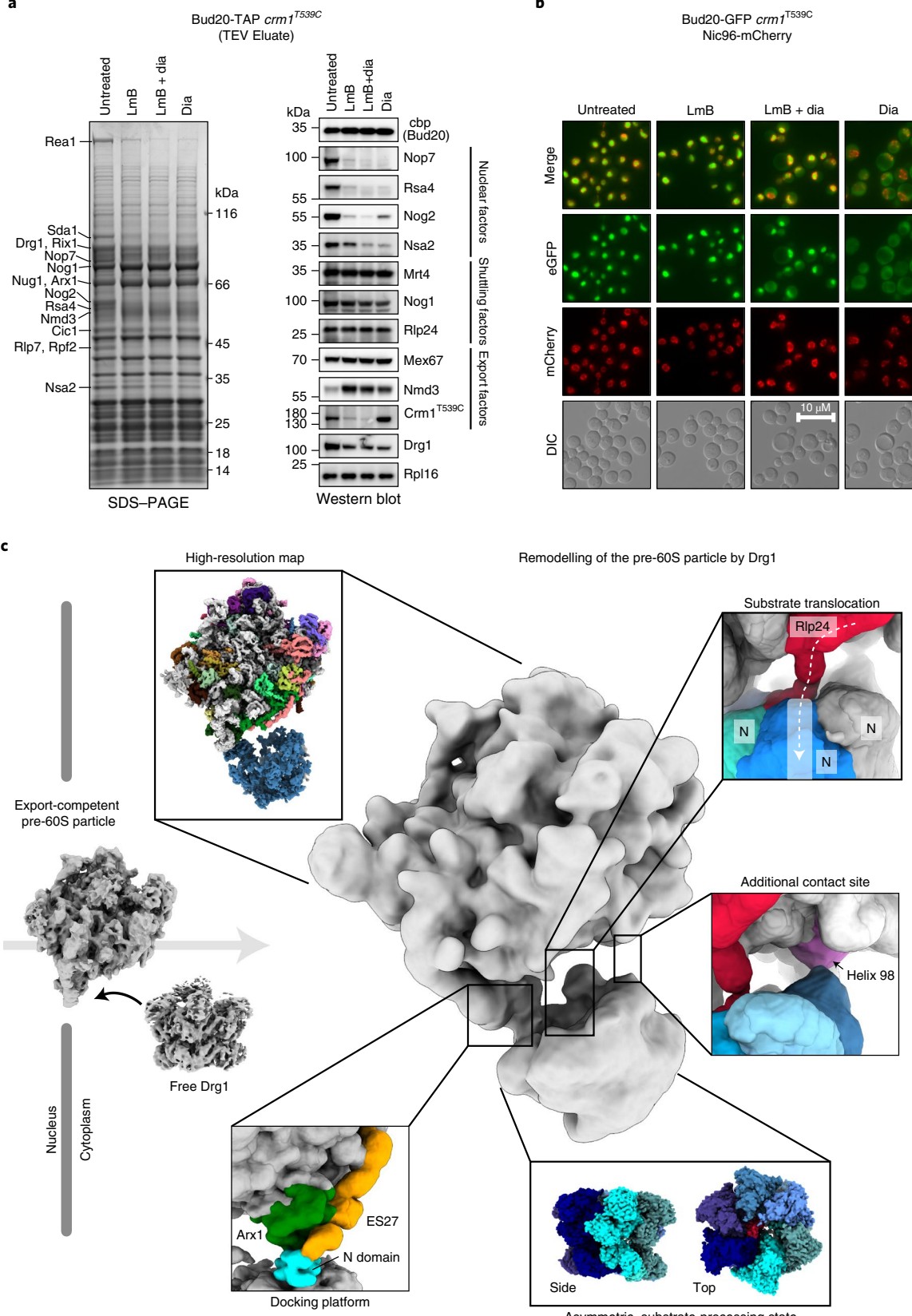

**Fig. 1 | Structure of the Drg1–pre-60S complex during extraction of Rlp24. a**, Bud20-TAP particles isolated from the LmB-sensitive *crm1*[T539C] strain (untreated) and after treatment with LmB and/or diazaborine (dia) were analyzed by SDS–PAGE and western blotting. To enable comparison, western blot signal exposures of all blots were adjusted to similar intensities to that of the untreated control. **b**, Localization of Bud20-GFP in the LmB-sensitive *crm1*[T539C] strain (nuclear membrane marker, Nic96-mCherry). **c**, Cryo-EM structure of the Drg1–pre60S complex. Drg1 binds via Arx1 and ES27 and additional, transient interaction sites (for example, 25S rRNA helix 98) to insert the Rlp24 C domain into its central pore for processive translocation.

**Table 1 | Cryo-EM data collection, refinement and validation statistics**

| | Structure of substrate bound DRG1 (AFG2) (EMD-14437) (PDB 7Z11) | Drg1 on pre-60S particle (EMD-14471) (PDB 7Z34) |
|---|---|---|
| **Data collection and processing** | | |
| Magnification | 81,000 | 81,000 |
| Voltage (kV) | 300 | 300 |
| Electron exposure (e⁻/Å²) | 60 | 60 |
| Defocus range (μm) | −0.5 to −2.5 | −0.5 to −2.5 |
| Pixel size (Å) | 1.07 | 1.07 |
| Symmetry imposed | $C_1$ | $C_1$ |
| Initial particle images (no.) | 3,148,330 | 3,645,306 |
| Final particle images (no.) | 114,728 | 48,536 |
| Map resolution (Å) | 3.2 | 4.2 |
| FSC threshold | 0.143 | 0.143 |
| Map resolution range (Å) | 4.2-3.2 | 8.5-4.2 |
| **Refinement** | | |
| Initial model used (PDB code) | 7NKU | 6RZZ, 6N8K, 6K8K, 4V6I, 3IZD |
| Model resolution (Å) | 1.7 | 3.3 |
| FSC threshold | 0.143 | 0.143 |
| Model resolution range (Å) | 10–1.7 | 10–3.3 |
| Map sharpening $B$ factor (Å²) | −73.2 | −80.4 |
| Model composition | | |
| Non-hydrogen atoms | 34,080 | 177,871 |
| Protein residues | 4,404 | 12,625 |
| Nucleotides | - | 3,625 |
| Ligands | AGS: 11 | UNK: 170 MG: 1 AGS: 11 |
| $B$ factors (Å²) (minimum/maximum/mean) | | |
| Protein | 36.86/137.92/70.04 | 25.24/338.17/138.45 |
| Ligand | 38.86/96.04/55.37 | 32.94/228.74/132.99 |
| R.m.s. deviations | | |
| Bond lengths (Å) | 0.004 (0) | 0.002 (0) |
| Bond angles (°) | 0.805 (38) | 0.485 (81) |
| **Validation** | | |
| MolProbity score | 1.89 | 2.49 |
| Clashscore | 8.64 | 16.17 |
| Poor rotamers (%) | 0.35 | 3.4 |
| Ramachandran plot | | |
| Favored (%) | 93.60 | 94.46 |
| Allowed (%) | 6.30 | 5.5 |
| Disallowed (%) | 0.09 | 0.03 |

levels and decrease of Nog2 demonstrate pronounced enrichment of export-competent particles (Fig. 1b). Residual levels of Drg1 in the preparation from LmB-treated cells likely arise from incomplete inhibition of Crm1/Xpo1, indicated by small amounts of the exportin. Furthermore, our pre-60S particles resemble the 'early cytoplasmic immediate (ECI) pre-ribosomal particle' purified after expressing a dominant negative Rlp24 allele[27]. As this mutant prevents Drg1 binding, this similarity confirms that our particles closely match the natural substrate of Drg1.

To capture Drg1 in the substrate-bound state, we used the slowly hydrolyzable ATP-analog ATPγS to prevent full extraction of Rlp24. This enabled us to identify Drg1-bound pre-60S particles and the AAA-ATPase alone (Fig. 1c). The structure of the complex reveals its unique geometry and visualizes initial recognition and subsequent translocation of Rlp24 during extraction. Our data identify the series of events, from initial binding of a AAA-ATPase to its macromolecular target up to substrate threading, and visualize major structural transitions during substrate translocation.

Our pre-ribosome structure shows defined densities for Arx1 at the exit of the PET and L12 and Mrt4 at the ribosomal stalk (Extended Data Fig. 2a,b). The center of the pre-60S particle is resolved to near-atomic resolution, and mostly allowed side-chain modeling. The particles lack Crm1/Xpo1 owing to LmB treatment (Fig. 1a). Because Nmd3 is already incorporated, the L1 stalk is in a closed position, with ribosomal protein L1 visible (Extended Data Fig. 2c,d). 25S rRNA helix 38 adopts a closed position and snaps down to the OB domain of Nmd3, as described recently[27,28]. The particles also contain the shuttling proteins Mrt4, Nog1, Tif6, Rlp24 and Bud20 (Fig. 1). The bait protein Bud20 is positioned right above Rlp24, explaining the concomitant release of both proteins immediately after export (Extended Data Fig. 2e,f)[15,29]. We also detect partial density (His20 to Lys74) for Nsa2 that forms two helices at the base of the P-stalk. (Extended Data Fig. 2g). The second helix of Nsa2 binds 25S rRNA helix 89 (Extended Data Fig. 2h) while, simultaneously, Mrt4 is contacted (Extended Data Fig. 2i). Thus, Nsa2 bridges the ribosomal stalk and helix 89, likely keeping the helix in a closed state. Furthermore, we observe weak density for Tma16 at the position reported in mammalian pre-60S particles[30].

**Recruitment of Drg1.** In the Drg1–pre-60S complex map, the pre-ribosome is well-defined, but we see only blurry density for the major part of Drg1. This indicates that the Drg1 hexamer is actively engaged and undergoes its conformational cycle. However, the primary interface between the AAA-ATPase and the particle is well-resolved. It is formed by the maturation factor Arx1 and the eukaryote-specific 25S rRNA expansion segment ES27, which are recognized by the same Drg1 N domain (Fig. 1c). Although, in most particles, Drg1 is solely anchored through Arx1 and ES27, three-dimensional (3D) variability analysis (3DVA) identified a minor population (approximately 2%; Fig. 1c) in which it additionally binds to 25S rRNA helix 98 and flanking regions. This contact is also visible in a subset of free Drg1 (Extended Data Fig. 1) and confirmed by crosslinking mass spectrometry (MS) showing linkages of Drg1 K13 and K24 with ribosomal protein L17 residues K13 and K180 that flank helix 98 (Supplementary Table 1).

The interactions with the pre-ribosome position Drg1 so that the Rlp24 C domain can be inserted into the central pore of the AAA-ATPase. The different structural states indicate that Drg1 dynamically associates with the pre-ribosome anchored via Arx1 and ES27 to ensure correct positioning and geometry for mechanical extraction of Rlp24.

**Arx1 and ES27 form a joint Drg1 docking platform.** We find tight contacts between Arx1 loops and the N domain of one Drg1 protomer (Fig. 2). The Drg1 N domains contain two distinct ß-barrel sub-structures ($N_N$ and $N_C$), with mainly negatively charged surfaces (Fig. 2a,b and Extended Data Fig. 3a). The subdomains are

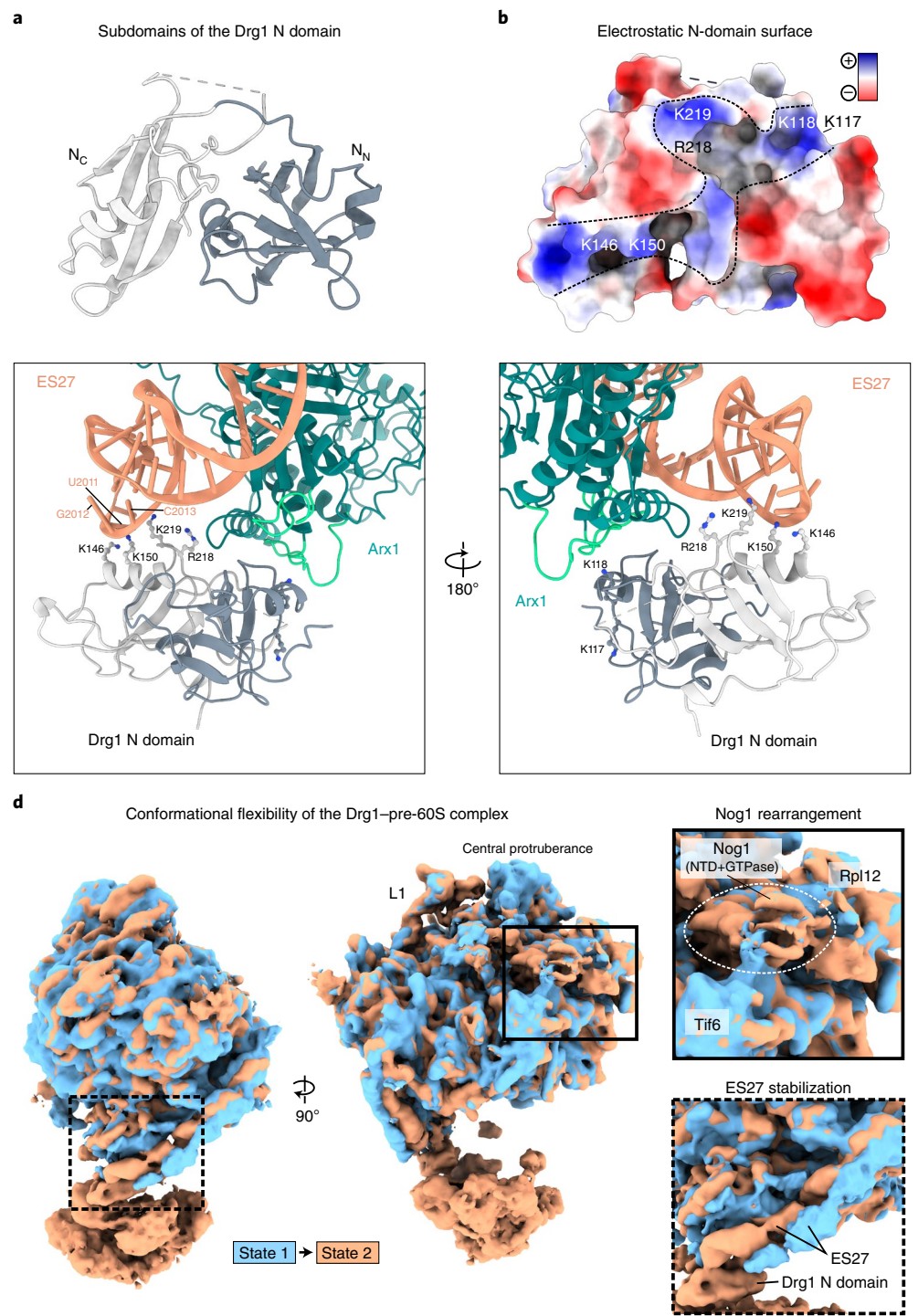

**Fig. 2 | Drg1 binds to a docking platform formed by Arx1 and ES27 of the 25S rRNA. a**, Bipartite subdomain organization of the Drg1 N domain. **b**, Electrostatic surface displaying a positively charged (blue) groove of the N-domain model. The binding cleft capturing Arx1 reaches from top to bottom, and ES27 interacts with a positively charged groove spanning both subdomains. **c**, The N domain of one Drg1 protomer binds Arx1 and ES27 simultaneously. Arx1 loop region 561 to 568 binds into the cleft between Drg1 $N_N$ and $N_C$. Additional contacts to Drg1 are formed by Arx1 loops 203–219 and 510–521 (light green). The bases U2010 to G2012 of ES27 form a single-stranded loop contacting lysine residues in the Drg1 N domain. **d**, 3DVA of the Drg1–pre-60S complex. Terminal frames of the 3DVA filtered to 7-Å resolution are superposed.

separated by a deep cleft in which the closely related AAA-ATPase p97 serves as adapter-protein-binding site[31].

In Drg1, the N-domain cleft is bound by the Arx1 loop region (561–568 aa) (Fig. 2c). 3DVA revealed a dynamic opening and closing of the cleft linked to the D2 nucleotide-loading state

(Supplementary Movie 1), which might capture Arx1 during recruitment of Drg1. Additional contacts are, for example, present between one Arx1 loop (510–521 aa) and the Drg1 $N_N$ subdomain loop (62–67 aa) as well as between a second Arx1 loop (203–219 aa) and the linker connecting the Drg1 $N_N$ and $N_C$ subdomains (118–127 aa).

The Drg1-Arx1 interaction is independent of ATP loading to the AAA-ATPase, as shown by GST pulldown (Extended Data Fig. 3b). This is consistent with interaction through the top of one single N domain, which does not require hexamerization. Therefore, the Walker B variant Drg1EQ1, which forms more stable hexamers[17], binds Arx1 similarly as the wild type does. As Arx1 does not affect in vitro ATPase activity of Drg1 (Extended Data Fig. 3c), it primarily serves as a structural adapter recruiting Drg1. A recruitment function of Arx1 is also suggested by lower levels of Drg1 on pre-ribosomes from an Arx1 deletion ($\Delta arx1$)strain (Extended Data Fig. 3d).

The same N domain that binds to Arx1 also interacts with the tip of ES27, which forms a single-stranded RNA loop exposing four nucleotide bases (Fig. 2c). Two of these bases (U2010 and U2011) are flipped toward Drg1. The ES27 loop is close to multiple lysines (117, 118, 146, 150 and 219) positioned in a groove in the N domain (Fig. 2b,c). The presumably dynamic interaction surface seems to be formed by a redundant set of amino acids. Single exchanges are tolerated, but exchanging multiple residues, including the predicted RNA-interacting residue K219, at the interface results in severe growth defects in the absence of Arx1 (Extended Data Fig. 3e). The tight contacts to the Drg1 N domain and additional contacts to Arx1 stabilize ES27, as the rRNA helix is better resolved when Drg1 is bound (Fig. 2d and Supplementary Movie 2). Taken together, the docking platform jointly formed by Arx1 and ES27 recruits Drg1 to the pre-60S particle, similarly to adapter proteins for other AAA-ATPases.

3DVA provided a dynamic view on the association of Drg1 with the pre-ribosome (Fig. 2d and Supplementary Movie 2). Upon recruitment, the N domain in the hexamer binding to Arx1 and ES27 undergoes a slight rotation, leading to remodeling and rotation of Arx1. Because the $N_C$ subdomain remains attached to ES27, this rotation is presumably transmitted to the 25S rRNA main body and functional sites of the pre-ribosome. In addition, 3DVA reveals high degrees of conformational flexibility on the pre-ribosome, for example in the area of the ribosomal stalk and L1 stalk, as well as a rotation of the Nog1 N-terminal helical bundle in the A-site (Fig. 2d and Supplementary Movie 2).

**The Drg1 N domains are involved in Rlp24 capture.** To determine the extraction mechanism of Rlp24, we searched for particle states depicting its interaction with Drg1. Indeed, we found a continuous density of Rlp24 entering the central channel of Drg1 (Fig. 1c). The Rlp24 C domain protruding from the pre-ribosome is contacted by the tips of four Drg1 N domains at the entry of the central pore. The free, substrate-engaged state of Drg1 confirms that the N domains engage with the inserted substrate (Fig. 3a).

The Drg1 $N_N$ subdomain is oriented toward the central pore of the hexamer. The very N terminus of Drg1 (aa 1–30, N-tip) contacts the substrate chain above the pore loops (Fig. 3b). In our ATPγS-bound state, four Drg1 N domains contact the substrate entering the pore. This number might be modulated by the nucleotide-loading state of Drg1. N-terminally-truncated Drg1 variants are functional (Fig. 3c and Extended Data Fig. 4) but exert a dominant negative growth phenotype when expressed as N-terminal GST fusion (Fig. 3d). Furthermore, the Drg1ΔN20 variant (which lacked amino acids 1–20) showed reduced stimulation of ATPase activity by the Rlp24C-domain, although the basal activity was like that of the wild type (Fig. 3e). Further truncations of the N-tip (Drg1ΔN28) prevented purification. Hence, the N-tips are not essential, but still influence the protein's functionality.

In contrast, the Rlp24 C domain is essential[12,13,16]. Deletion of the last 16 Rlp24 residues (Fig. 3f, Δtail) prohibits binding of Drg1 (Fig. 3g), is lethal and exerts a dominant negative growth defect (Fig. 3h). Moreover, it abolishes stimulation of the ATPase activity of Drg1 (Fig. 3i). Thus, the Rlp24 tail is crucial for interaction and

stimulation of Drg1. This is supported by the finding that exchanging three positively charged residues in the Rlp24 tail (R191E, K195E and K196E; designated RKK>E) reduces binding and stimulation of Drg1 (Fig. 3g,i). This suggests that recognition and activation of Drg1 by Rlp24 involve sequence-specific features of the Rlp24 C domain.

The presumably transient interaction between the Drg1 N-tip and Rlp24 C domain is confirmed by crosslinking MS, which links Drg1 K13 and K24 to K184 in the Rlp24 C domain (Fig. 3f and Supplementary Table 1).

**High-resolution structure of Drg1 substrate threading.** Anchored through Arx1 and ES27 at the pre-ribosome, Drg1 shows a staircase-like conformation that is prototypical for substrate-engaged AAA-ATPases[32–34]. Thus, we captured Drg1 mid-processing after initial recognition of the substrate. For better local resolution, we analyzed free Drg1 hexamers present in our sample.

Our dataset contains both a symmetric and an asymmetric conformation of free Drg1 (Fig. 4). While the N and D1 rings of both forms are nearly superposable, the D2 ring is rotated by about 30° clockwise in the asymmetric structure (Fig. 4a and Supplementary Movie 1). The symmetric conformation is characteristic of AAA-ATPases not engaged in substrate processing[33]. Consistently, additional density is found at the entrance of the central channel in contact with the N domains, but no density is found inside. The symmetric structure is highly similar to Drg1 in complex with the inhibitor diazaborine that blocks ATP hydrolysis in D2 (refs. [16,17,20]). This suggests that diazaborine fixes Drg1 in the symmetric form and prevents conversion to the active conformation.

The free asymmetric Drg1 hexamers in a staircase-like arrangement are highly similar to the particle-bound state, but with much higher local resolution (up to 3.2 Å). Intriguingly, the pore harbors a polypeptide chain engaged by pore loops of several protomers, confirming that this is the active substrate-processing state (Fig. 4). The free substrate-engaged Drg1 hexamer was fitted into the cryo-EM map of the Drg1–pre-60S complex (Fig. 4b). In both cases, at least four protomers contact the substrate; one is less well-resolved and represents the detached seam protomer. The protomers were designated clockwise in top view from A to F, starting with the protomer exhibiting the D1 pore loop in the highest position (Fig. 4c). Protomer D, positioned counterclockwise to the seam protomer E, binds Arx1 and ES27 through its N domain. However, the seam is presumably rotating from protomer to protomer, and binding to the particle influences the transition kinetics, leading to enrichment of the seam adjacent to the anchoring protomer in our structure.

The polypeptide chain inside the pore revealed the substrate interactions with the pore loops in both AAA domains (Fig. 4c,d). ATPγS locked Drg1 in a mid-processing state and prevented full translocation. All nucleotide-binding domains are loaded with ATPγS, except the D2 domain in the seam protomer E, which lacks a nucleotide, similar to related AAA-ATPases[32,33]. Four protomers (A, B, C and D; Fig. 4) directly contact the substrate through their conserved D1 pore loops. The arrangement pattern of the pore loops reflects substrate binding of the N-domains (see above). Y319 of the D1 pore loop in protomer A is in the top position, with Y319 of protomers B, C and D contacting the substrate chain in increments of two amino acid residues below. This indicates that Drg1 utilizes a hand-over-hand grabbing mechanism to gradually translocate the substrate[32,33]. The pore loops of the seam protomer E and the D1 pore loop of protomer F are not binding the substrate. In D2, in contrast, protomer F already contacts the substrate and the pore loop of protomer E is detached (Fig. 4c,d). Thus the ATPγS-bound state demonstrates that the D1 and D2 domains are highly cooperative but are out of phase during substrate processing.

Except for protomer A, all D2 domains directly contacting the substrate expose their intersubunit signaling (ISS) motif into the

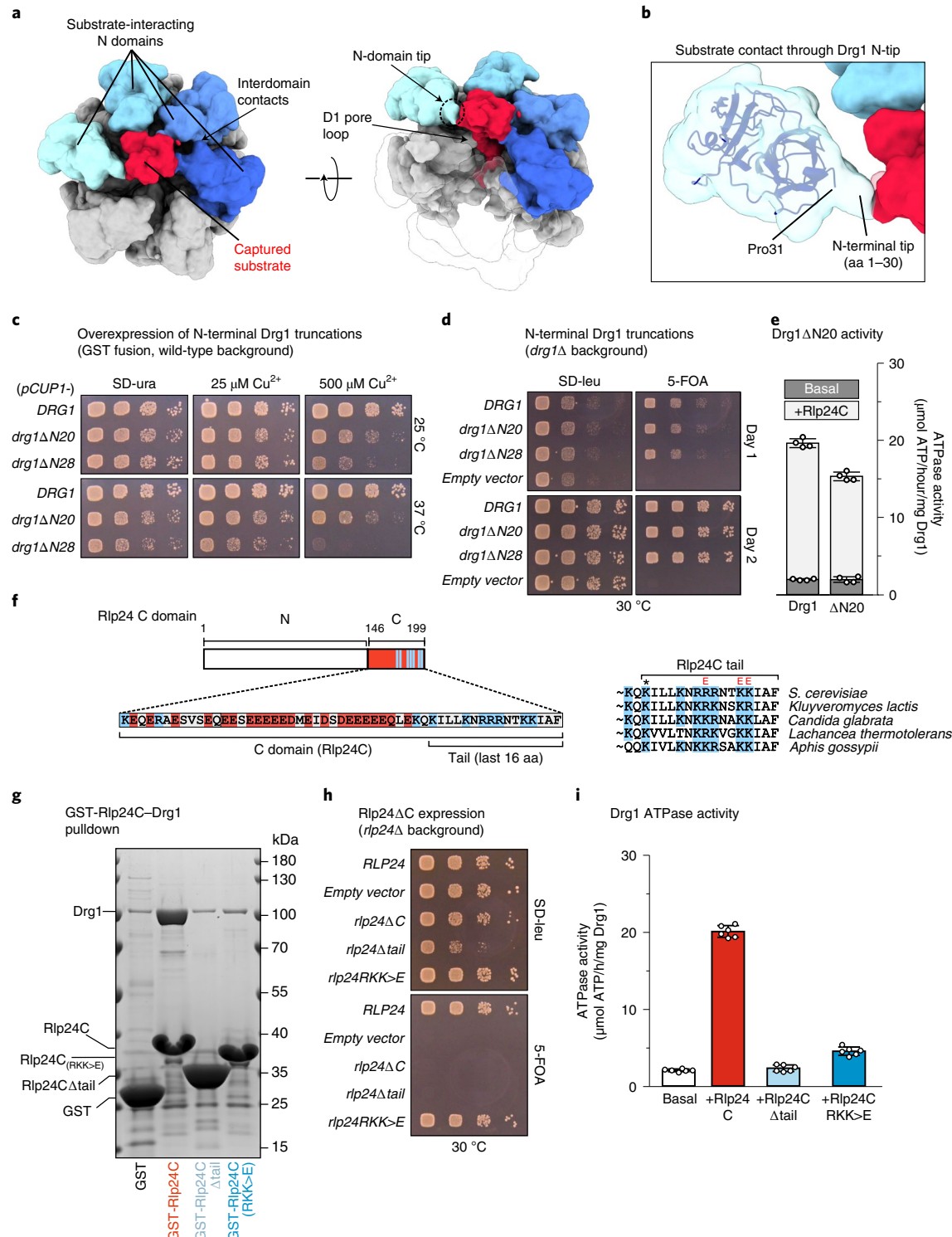

**Fig. 3 | Coordinated substrate recognition through N domains. a**, Drg1 hexamer map in top view at low threshold (0.05). **b**, The Drg1 N-terminus (aa 1–30, N-tip) is not well-resolved but likely represents the density contacting the substrate chain in the center of the hexamer. **c**, N-terminally-truncated Drg1 variants were overexpressed as GST fusions under control of the $Cu^{2+}$-inducible *CUP1* promoter in the wild-type background. **d**, Expression of N-terminally-truncated Drg1 variants lacking amino acids 1–20 (ΔN20) or 1-28 (ΔN28) from their endogenous promoter in a Δ*drg1* shuffle strain. **e**, ATPase activity of N-terminally-truncated Drg1 (ΔN20), with and without Rlp24C. Mean and s.d. of two biological replicates, each measured in duplicate ($n=4$). **f**, Acidic stretch (red) in the Rlp24 C domain (aa 146–199), followed by conserved alkaline residues (blue). The last 16 amino acids are designated as the C-terminal tail. K184 (*) crosslinks to Drg1 K13 and K24. 'E' marks residues exchanged to glutamate (R191E, K195E and K196E, together designated as RKK > E). **g**, Binding of Drg1 to GST-Rlp24C (aa 146–199) variants (GST pulldown). **h**, Spot assay of a *rlp24*Δ shuffle Rlp24 strain expressing either Rlp24ΔC (deletion of the whole Rlp24 C domain) or Rlp24Δtail (deletion of the last 16 residues). A strain expressing fulllength Rlp24 (*RLP24*) serves as control. Dominant negative growth is indicated by reduced growth on SD-leu (yeast synthetic drop-out medium lacking leucine). Non-functional constructs prohibit growth on 5-FOA. **i**, Drg1 ATPase activity in the presence of Rlp24 variants. Mean and s.d. of two biological replicates, each measured in triplicate ($n=6$).

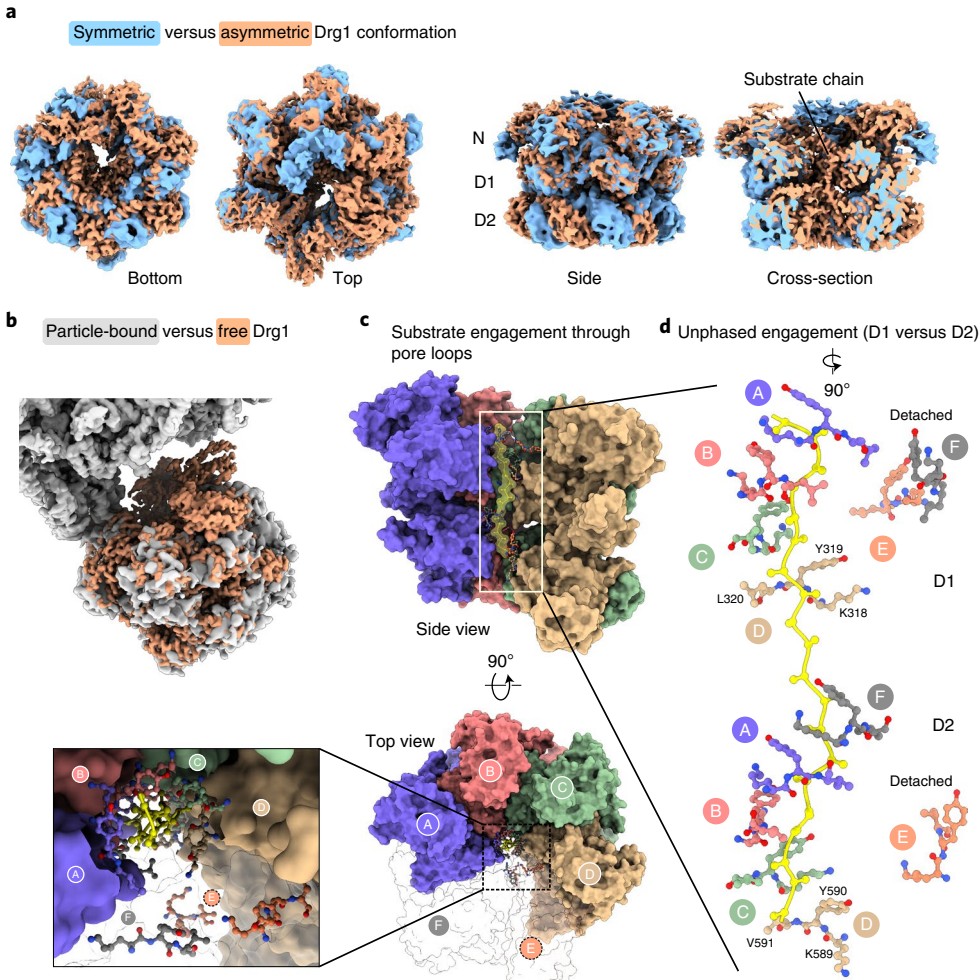

**Fig. 4 | Substrate-threading mechanism of Drg1. a,** Superposition of symmetric (blue) and asymmetric (orange) maps of Drg1. **b,** 3.2-Å asymmetric free Drg1 map (orange) versus particle-bound Drg1 map (gray). **c,** Substrate translocation (top and side view). Substrate (poly-alanine chain) and Drg1 pore loop residues shown as sticks, and the rest of the hexamer model is shown as surface. **d,** Unphased pore loop substrate engagement of D1 and D2. The conserved tyrosine (Y319 in D1 and Y590 in D2) plus adjacent residues of the pore loop motif are shown.

adjacent nucleotide-binding site (Extended Data Fig. 5). A conserved aspartate (D645) in the ISS loop interacts with the arginine finger of the same protomer[35–37]. The ISS can transform into a triangular loop that coordinates the protomers by preventing premature ATP hydrolysis in the adjacent protomer. In the D2 domain of substrate-engaged Drg1, the ISS loops of protomers B, C and D are inserted into the nucleotide-binding pocket of protomers A, B and C, respectively. The protomer A ISS is partially retracted from protomer F. However, the pore loop is still engaged in substrate binding, indicating that this protomer is in a transition state prior to ATP hydrolysis in D2. In E and F, the ISS loops are fully retracted. There is a strict correlation between ISS insertion and definition of the D1D2 linker of the adjacent protomer. This is explained by interaction of residues in the center of the loop with the D1D2 linker (Extended Data Fig. 5 and Supplementary Movie 1). The ISS is further stabilized by interaction with R606 at the end of the pore loop helix. This interaction thus links substrate association with the state of the ISS and might ensure that ATP hydrolysis occurs only at the right stage of the extraction cycle.

**Substrate extraction through hand-over-hand translocation.** 3DVA of substrate-engaged Drg1 visualized the interplay of individual domains during substrate translocation. This revealed a chain of events linking the nucleotide-loading state in D2 with grabbing and releasing the substrate by the pore loops and N domains. This enables stepwise passing along of the substrate during translocation. The experimental data visualizing the major stages, from substrate recognition to translocation (Fig. 5a), are shown in Supplementary Movie 1, with emphasis on the hand-over-hand translocation mechanism (Fig. 5b). The structural basis of substrate translocation is described in detail below.

The perpetuated conformational changes start in D2 of the seam protomer that we captured in the apo- and nucleotide-bound state. In the starting frame of the side view, the current seam protomer (P) still has the nucleotide bound in D2, albeit there is no contact to the substrate. This presumably reflects a transition state prior to repositioning of this protomer. D2 of the clockwise adjacent protomer (P+1) is, at this stage, also not contacting the substrate.

Upon releasing the nucleotide from the seam D2, the whole D2 domain is lifted. The rise of the helical subdomain elevates the α/β subdomain of the clockwise (P+1) D2 domain, and the D2 pore loop of this domain moves toward the substrate, providing the next grip in top position of all D2 pore loops. At this stage, the seam is still detached, and thus has the highest degree of freedom to move. The counterclockwise P−1 D2 domain is now in the lowest position and thus the next in line to take the current seam's place. In the

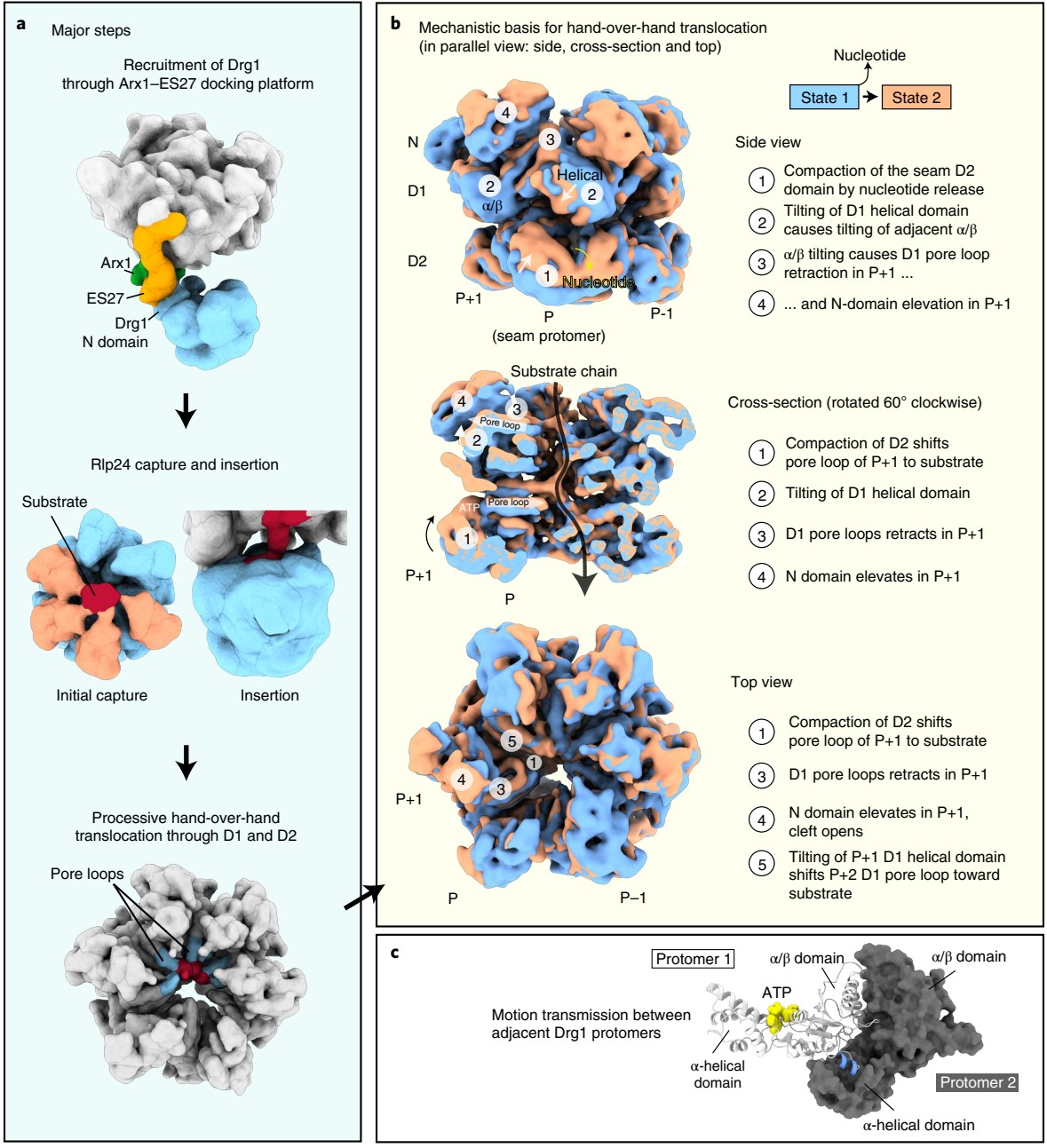

**Fig. 5 | Mechanism of substrate extraction and translocation by Drg1. a**, Major steps of Rlp24 extraction: recruitment of Drg1 to the particle, initial substrate capture, insertion into the pore and processive translocation. Filtered maps are shown for simplification. **b**, Structural basis for substrate translocation. The same conformational changes are shown from three different views in parallel: side, top and cross-section. 'P' denotes the seam protomer. Nucleotide loading in the current seam protomer transmits conformational changes to the adjacent protomers (P+1 and P+2) (see also Supplementary Movie 1). Two terminal frames from the 3DVA (7-Å filter resolution) are shown. **c**, Motion transmission between adjacent protomers occurs through the helical subdomain to the next clockwise AAA domain, as exemplified for D1. A short helix (residues 269–273, blue) of the α/β domain is hooked into the helical domain of the adjacent protomer, which coordinates movements of both domains. The nucleotide (yellow) is positioned between the two subdomains.

starting frame, nucleotide hydrolysis in the P−1 D2 is still blocked by the inserted ISS, presumably to prevent premature hydrolysis. Nucleotide dissociation in the seam D2 fully retracts the arginine finger from the nucleotide in the P−1 site and partially retracts the ISS (Supplementary Movie 1).

The described movements in the D2 ring are transmitted to the D1 ring. The lifting of the whole seam D2 domain and the rightward movement of the α/β subdomain converts into an inward rotation and leftward translocation of the helical subdomain of D1, mediated through the D1D2 linker. At this stage, the conformational change

is also transmitted to the adjacent (P+1) D1 domain, because the helical subdomain of the seam tilts the D1 α/β subdomain of protomer F (Fig. 5b). This has two important consequences: First, it retracts the D1 pore loop helix of protomer F from the substrate. Second, it pushes the N domain of protomer F upward. Elevation of the N domain is thereby driven by the N–D1 linker connected with the $N_C$ subdomain and through interaction of the D1 loop around Y337 with the $N_N$ subdomain. Since $N_N$ and $N_C$ are individually anchored via these two interactions, the upward movement opens the cleft between the subdomains. In a global view, the D1 domains

of P+1 and the P+2 α/β subdomain spiral upward in the clockwise direction, and the P+2 D1 pore loop and N domain move toward the substrate. Thus, the D1 and D2 rings act out of phase, with the D2 ring getting the next hold on the substrate one step before the D1 ring.

The structural changes in the D2 domain of the seam protomer cause leftward movement of the D1 helical domain, which is why the D1 and D2 rings are rotated by ~30° against each other. This pulls down the D1 α/β domain and the associated N domain of the seam protomer and causes rotation of the helical domain of protomer C, which affects positioning of the N domain of D. Thus, the ATP-loading state of the seam protomer also modulates N-domain substrate engagement in protomer D. As Drg1 is bound through the N domain of protomer D, positioning of the AAA-ATPase relative to the pre-ribosome will change during the cycle of ATP binding and hydrolysis, explaining its blurry appearance in our particle-bound structure.

In summary, nucleotide dissociation in the seam D2 domain initiates a cascade of conformational changes that remotely triggers substrate engagement in the adjacent protomers. Consecutive grabbing and releasing of the substrate by the pore loops creates directionality and pulls the substrate chain through the pore in a stepwise manner.

**Quantification of Rlp24 release from pre-60S particles.** To assess the mechanism of recognition, we quantified binding of Drg1 to the pre-60S particle using surface plasmon resonance (SPR, Fig. 6). We captured purified pre-60S particles through the Bud20-TAP tag on an SPR chip and injected increasing concentrations of purified Drg1 in the presence of ATPγS (Fig. 6a). Additionally, we measured binding of Drg1 to the immobilized Rlp24 C domain (Fig. 6b). Binding of Drg1 to Rlp24C follows a hyperbolic behavior with a dissociation constant ($K_D$) of 2.7 μM (Fig. 6c). Binding of Drg1 to the pre-ribosome could not be described by a hyperbolic curve but follows a sigmoidal dose–response curve. This revealed a half-maximal effective concentration ($EC_{50}$) of 46 nM and a Hill coefficient of 3.5 ± 0.9, indicating positive cooperativity.

The same setup was used to investigate substrate release. Since Bud20 is localized on top of Rlp24 (refs. [10,27]) (Extended Data Fig. 2f), it is liberated from the pre-ribosome concomitantly with Rlp24 (refs. [15,38]). This enabled us to measure release of the pre-ribosome from chip-bound Bud20-TAP and thereby quantify substrate processing by Drg1. We injected Drg1 in the presence of 1 mM ATP and measured the release of the particles from the sensor chip. With 37.5 nM Drg1 and ATP, we observed a linear decrease (~1 RU/second) of chip-bound particles (Fig. 6d). This strictly depends on ATP hydrolysis, because no release was measured using ATPγS (Fig. 6d). In addition, diazaborine, which specifically targets the D2 domain of Drg1 (refs. [16,17]), fully blocked the release in the presence of ATP. However, lower RU levels were reached as compared with ATPγS, suggesting that the drug interfered with pre-ribosome binding (Figs. 1a and 6d). Taken together, the results indicate that Drg1 is highly efficient in releasing Rlp24 (and Bud20) from pre-60S particles and the release strictly depends on ATP hydrolysis in the D2 domain.

## Discussion

**Impact of Drg1 activity on downstream maturation.** Rlp24 extraction simultaneously liberates Bud20, which shows extensive contacts with Rlp24. Only then can the ribosomal protein L24 be incorporated. Owing to the tight entanglement with Nog1, Rlp24 extraction presumably leads to at least partial release of Nog1, as proposed previously[39]. Rlp24 thereby might act as a one-sided lever that facilitates extraction of the Nog1 C terminus from the PET. Indeed, at least partial extraction of Nog1 is suggested by chemical crosslinking of Drg1 K13 and Nog1 K607 (Supplementary Table 1). K607 resides inside the PET and might not be accessible as long as the Nog1 C terminus is fully inserted. Moreover, 3DVA indicates that the rotation of ES27 upon Drg1 binding is transmitted through the 25S rRNA body to the whole particle up to the Nog1 N-terminal domain. ES27 functioning as a physical linker between distant sites of the ribosome was already suggested in previous studies[40,41]. 3DVA also revealed rotation of the GTPase domain and the helix bundle inserted in the A-site, destabilizing Nog1. Thus, Drg1 not only pulls off Rlp24 from the pre-ribosome, but might restructure and prime the particle for downstream maturation. However, we cannot resolve whether these changes are directly caused by binding of Drg1 or reflect intrinsic flexibility of the pre-ribosome, but they provide an interesting starting point for future studies.

**RNA and protein contacts position Drg1 at the pre-ribosome.** The joint Arx1 and ES27 docking platform recruits Drg1 near the PET exit. Although the N domains of type-II AAA-ATPases are known to mediate substrate and co-factor interactions[31], this is to our knowledge the first time that the N domain shows specific contacts to RNA.

The unique geometry of the complex allows further insights into biomechanical aspects of the release process. Anchoring through Arx1 and ES27 may establish the right distance and angle for Rlp24 extraction. The essential recognition of the Rlp24 C domain and additional contacts of Drg1 with helix 98 might explain why Arx1 is still dispensable for growth. Nevertheless, Arx1 enhances recruitment of Drg1 (Extended Data Fig. 3d) and hence the efficiency of extraction.

Drg1 hangs freely on a single protomer and is not close to the particle's surface, which has mechanistic implications on the forces it may generate. A direct pulling force often discussed for AAA proteins is expected to reorient the particle and stabilize the complex, leading to better visibility of the hexamer. In contrast, in our ATPγS structure, we see an entirely flexible binding mode that will support the generation of entropic forces, in line with molecular machine theoretical considerations[42]. This would, rather, be in line with the commonly proposed Brownian ratchet model than a power stroke model. Particle averages indeed show that Drg1 can move relative to the 60S body, which may generate an entropic pulling force, loosening Rlp24. We speculate that anchoring of Drg1 is necessary to generate this force while AAA-ATPases normally merely entrap substrates thermodynamically. For the ATP-dependent Hsp70 chaperones, a unifying entropic pulling

**Fig. 6 | Quantifying pre-ribosomal particle recognition and release by Drg1. a**, Binding of Drg1 to pre-60S particles captured on an SPR chip. pre-60S particles purified after LmB treatment were captured via the Protein A moiety of the Bud20-TAP tag on IgGs immobilized on the sensor chip. We injected 25–150 nM of purified Drg1 for 3 minutes in the presence of ATPγS. Sensorgrams of one representative binding series are shown. RU, response units. **b**, Binding of Drg1 to the Rlp24 C domain. GST-Rlp24C was immobilized as a ligand, and 0.075–10 μM of purified Drg1 was injected for 3 minutes in the presence of ATPγS. Sensorgrams of one representative binding series are shown. **c**, Normalized RU plotted over Drg1 concentration. Binding to Rlp24C was fitted using a one-site binding hyperbola. Binding to the Bud20-TAP particle was fitted using a sigmoidal dose–response curve (semi-log scale). Three biological replicates (n = 3) are shown. $EC_{50}$ or $K_D$ are depicted with the 95% confidence interval (CI), hill slope with s.e. **d**, Measuring substrate processing using SPR (particle-release assay). Pre-60S particles purified after LmB treatment were captured on a sensor chip through their Bud20-TAP tag. Particle release triggered by Rlp24 extraction was quantified by measuring the RU over time in the presence of ATP, ATP + diazaborine (Dia) or ATPγS (binding control) with increasing Drg1 concentrations (37.5–150 nM). The release rate ($\Delta RU/\Delta t$) was determined from the linear ranges of four 37.5 nM Drg1 injections (n = 4). Sensorgrams of one representative release series are shown.

mechanism of substrate translocation was proposed in addition to the commonly discussed power stroke and Brownian ratchet models[43,44]. However, currently we cannot derive the full mechanism behind the release of Rlp24, and the described models may not be mutually exclusive[45].

**Substrate recognition and handling by the Drg1 N domain.** In our ATPγS-bound structure, four N domains simultaneously encounter the substrate above the central channel in a spiral and direct it into the pore. The N domains show multiple variable contacts to the substrate, indicating either several binding sites or pronounced

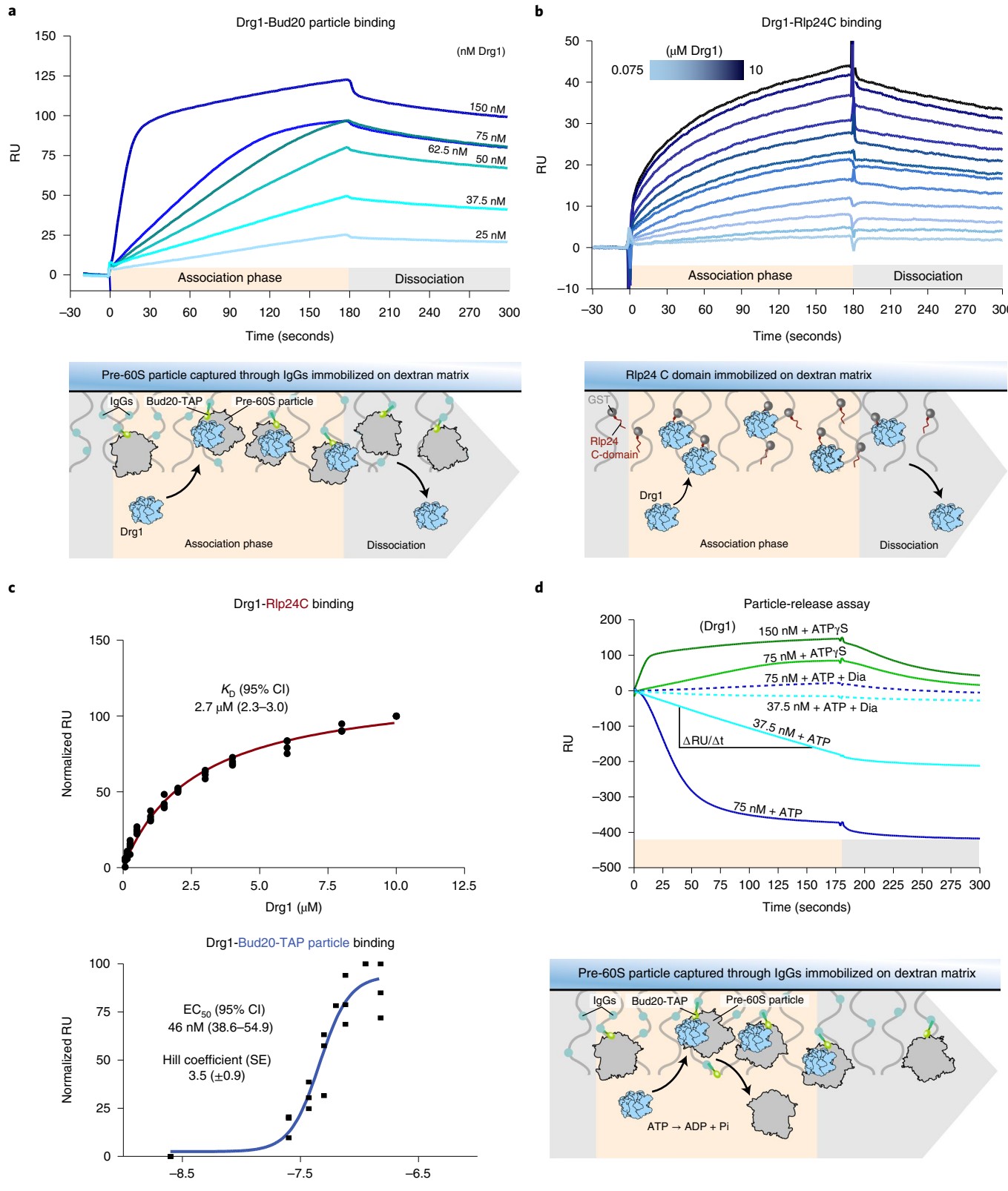

flexibility. Additional interactions occur between N domains of adjacent protomers, which could be important for coordination of the protomers during extraction. Because the N-tip crosslinks to the Rlp24 tail, this might facilitate initial insertion of the C domain before the D1 pore loops take over. We cannot, however, definitely assign the involvement of the N tips temporally and distinguish between a role in the initial insertion or in later stages, for example to prevent backtracking of the translocated polypeptide chain. In any case, deletion of the first 20 N-terminal Drg1 residues reduces stimulation of ATP hydrolysis by Rlp24 underlining their importance for substrate interaction.

**Processive translocation of Rlp24.** Extraction of Rlp24 is driven by interactions with the pore loops of both Drg1 AAA domains and thus follows a conserved scheme in type-II AAA-ATPases[46–48], reviewed in refs. [32–34]. Indeed, superposition of the substrate-bound states of Cdc48 and Drg1 shows that the substrate chains adopt an almost identical corkscrew-like arrangement, and the contacts with the pore loops are nearly superposable (Extended Data Fig. 5e,f). By consecutively grabbing the substrate after two residues, the protomers jointly translocate the substrate in a processive manner. The two ATPase rings work together in a synchronized, albeit slightly unphased, manner. We find four (D1) and five (D2) substrate-interacting Drg1 protomers, respectively, whereas in Cdc48 (PDB: 6OPC)[46] or VAT (PDB: 5VCA)[48] structures five protomers are associated in both domains. Thus, we captured an intermediate state in which the AAA domains of one protomer (protomer F in our structure) are asynchronous (Fig. 4). The unphased behavior of the D1 and D2 rings likely guarantees unidirectional force generation for pulling off Rlp24 and counteracts backtracking.

Consistent with our previous findings[12], 3DVA indicates that the D2 domain is the main determinant for extraction. It modulates positioning of all major elements of the AAA-ATPase for substrate handling in dependency of nucleotide loading. Upon nucleotide dissociation in D2, movements and forces spread between protomers seem to be mainly transmitted via the helical subdomains of the AAA module. The helical subdomains act as rigid bodies and transfer rotational movements, for example through interaction with a short helix (aa 269–273 in D1 and 539–545 in D2) from the α/β subdomain of the adjacent protomer (Fig. 4c). The α/β subdomain then transfers pulling and torsion forces to the N domain and the pore loop helices. This modulates substrate engagement through the pore loops and positioning of the N domain. The allosteric p97 inhibitor MNS-873 supports the importance of the helical-α/β subdomain interface for subunit communication of AAA-ATPases. MNS-873 binds into the groove between the short helix and the ISS motif that modulates ATP hydrolysis in adjacent sites in dependency of pore loop substrate interaction[37]. MNS-873 therefore traps p97 in an inactive state and prevents substrate engagement.

The coordinated movements visualized in our 3DVA can take place only when the D1 subdomains form a stiff unit, which likely requires a bound nucleotide between the helical and α/β subdomains. This explains why Walker A mutations preventing nucleotide binding in D1 are deleterious for AAA-ATPases. The movements upon nucleotide loading in D2 cover substrate engagement of the pore loops of the D1 and D2 rings and modulation of the N domains, including its substrate grabbing and pushing toward D1. Thus, nucleotide hydrolysis in D1 is presumably not required for these transitions, which is in line with D1 Walker B mutants of Drg1 being viable[12,16].

With our new SPR setup, we could measure, in real-time, both binding of Drg1 to the pre-ribosome in the presence of ATPγS and ATP-dependent extraction of Rlp24 through the concomitant release of Bud20-TAP. The measured release of 1 RU/second corresponds to ~1.5 pre-ribosomal particles per Drg1 hexamer per second or ~90 particles per minute (Methods). Thus, only ~20 Drg1

hexamers are sufficient to drive maturation of 2,000 pre-ribosomes formed per minute in a yeast cell[49]. This number reasonably agrees with the estimated ~130 Drg1 hexamers (~800 molecules) per yeast cell[50].

Taken together, our data visualize the assembly of a AAA-ATPase on a large and complex target structure and dissect the chain of events during mechanical extraction of its specific substrate. This provides an important step forward to understand the dynamics and conserved modus operandi of these sophisticated biomechanical devices.

## Online content

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

## Methods

**Materials availability.** Plasmids and strains generated in this study are available from the corresponding authors upon request. Materials, chemicals and reagents are listed in Supplementary Table 2.

**Strains and plasmids.** Yeast and bacterial strains are listed in Supplementary Table 2. Growth/culture conditions for the respective experiments are described in the method details.

**Drug treatment of GFP/mCherry-tagged strains.** The *Saccharomyces cerevisiae* leptomycin sensitive (crm1[T539C]) mutant strain chromosomally expressing Bud20-GFP and the mCherry-tagged nuclear membrane protein Nic96 was grown to $OD_{600}$ 0.6 in SD medium supplemented with all amino acids. Thereafter, LmB (135 ng/μl) and/or diazaborine (10 μg/ml) were added, and cells were inspected by fluorescence microscopy after 30 minutes of treatment. As negative control, untreated cells were inspected. A Leica DM6 equipped with a ×100 HC PL APO (1.4) objective was used for fluorescence imaging.

**Pre-ribosomal particle purification for cryo-EM and SPR.** Pre-60S particles were isolated from a LmB-sensitive *S. cerevisiae* mutant strain (crm1[T539C]) using Bud20-TAP as bait. Cells were grown to late log-phase ($OD_{600}$ 1–1.2) and then treated with 135 ng/μl μM LmB for 30 minutes. Cells were collected by centrifugation (4,500g, 1 minute, 25 °C) and stored at −20 °C. Affinity purification via the Tandem affinity purification (TAP) tag was performed as described[38,51,52]: crude extracts were prepared by breaking the cells in a Merkenschlager bead mill in the presence of 0.6 mm glass beads (4 minutes with $CO_2$ cooling every 30 seconds) in buffer A (20 mM HEPES-NaOH, pH 7.5, 10 mM KCl, 2.5 mM $MgCl_2$, 1 mM EGTA, 1 mM DTT, 0.5 mM PMSF and protease-inhibitor-mix FY (Serva)). The homogenate was centrifuged at 40,000g, 4 °C, 30 minutes. For cryo-EM analysis, export-competent particles were purified using homemade magnetic beads containing covalently linked rabbit IgG and subsequent TEV protease cleavage[38]. For SPR analysis, particles were purified on calmodulin resin using the calmodulin binding peptide of the TAP tag and subsequently eluted with EGTA. The particle concentration of the eluate was estimated by $OD_{260}$ measurement of the RNA content in the sample.

**Drg1 purification.** GST-tagged wild-type Drg1 and the EQ1 (E346Q) variant were overexpressed in yeast as described[12,16]: the expression strain was inoculated to a starting $OD_{600}$ of 0.01, incubated at 30 °C at 110 r.p.m. in baffled flasks. Protein expression was induced by immediate addition of 0.025 μM $CuSO_4$, and cells were collected after 24 hours. Affinity purification was performed as described[12,16–18]: frozen cells were thawed in lysis buffer (50 mM Tris-HCl, 150 mM NaCl, pH 7.4, 1 mM DTT, 1× complete protease inhibitor cocktail (Roche) and 0.5 mM PMSF) and disrupted by vigorous shaking with 0.6 mm glass beads in a beadmill (Merckenschlager). Homogenates were centrifuged twice at 40,000g at 4 °C for 15 and 30 minutes, respectively. Crude extracts were incubated for 90 minutes at 4 °C with GSH-agarose beads (Sigma Aldrich) for affinity purification of GST-tagged Drg1. After consecutive washing steps (3× with lysis buffer plus 1 mM EDTA, 1 mM DTT and 1× with elution buffer plus 1 mM DTT), the protein was eluted. For elution, the GST tag was cleaved off using Prescission protease (GE Healthcare/Cytiva) overnight at 4 °C on a rotator in elution buffer suitable for the respective experiment. Protein concentration was measured using the Bradford assay (Biorad) with BSA calibration curve.

**Assembly of the Drg1–pre-60S particle complex for cryo-EM.** For reconstruction of the Drg1–pre-60S complex, purified Drg1 was eluted in cryo-elution buffer (20 mM HEPES-KOH, 150 mM potassium acetate, 5 mM magnesium acetate, 0.005% Tween-20, 1 mM DTT, pH 7.6), adjusted to 1.75 mg/ml (12 μM) and mixed in a 1:1 ratio with 200 nM export-competent pre-60S particles in the presence of 2 mM ATPγS and mixed for 10 minutes prior to plunge freezing. Quantifoil (R1.2/1.3) copper grids were glow-discharged for 60 seconds using an ELMO glow discharge system. Grids were loaded with 4 μl of the Drg1–pre-60S mixture and plunge frozen in liquid ethane using an FEI Vitrobot Mark IV at 4 °C and an environment with 100% humidity. The blotting force was set to 6, along with a blotting time of 7 seconds.

**Cryo-EM imaging settings.** Cryo-EM data for the Drg1–pre-60S complex were collected on a FEI Titan Krios G3i in conjunction with a Gatan K3 BioQuantum direct electron detector using a slit width of 20 eV. The camera was operated in counting mode using hardware binning (pixel size of 1.07 Å pixel⁻¹) and dose fractionation. One movie contains 54 images resulting in a total dose of 60 e⁻/Å² with a total exposure time of 4.84 seconds. The microscope was operated at 300 kV in nanoProbe energy-filtered transmission electron microscopy (EFTEM) mode at a nominal magnification of ×81,000. The dataset was acquired using SerialEM[53] with an active beam tilt and astigmatism compensation. The acquisition scheme was designed to collect nine holes once per stage movement. Since the resulting maps showed preferred orientation, an additional, 34°-tilted dataset was collected with the same settings. The acquisition scheme was adapted to three holes per stage movement along the tilting axes. The required tilt angle was calculated with Relion v3.0.

**Image processing.** Image processing was mostly performed in Cryosparc v3.0 (ref. [54]). Micrograph preprocessing included motion correction (patch motion correction) and CTF determination (patch CTF estimation). Micrographs and power spectra were individually inspected using the manually curate exposures function to exclude low quality micrographs due to ice contamination, devitrification or strong drift.

For the Drg1–pre-60S complex, 6,214 high-quality micrographs from the untilted dataset as well as 4,221 micrographs from the tilted dataset were used for particle picking. Initial 2D class averages generated from manually picked particles of the Drg1–pre-60S complex were used for template picking in both datasets. Selected 2D classes from both datasets were combined, resulting in an initial set of 3,645,306 particles. After multiple rounds of 2D classification, 1,782,014 particles were used to generate an initial ab initio 3D model. Subsequently, multiple rounds of heterogeneous refinement, as well as 3DVA followed by clustering, were performed to separate pre-ribosomal particle populations. On the basis of the conformation of the L1 stalk, export-competent Bud20-TAP particles (closed L1) were differentiated from earlier Bud20-TAP populations or mature 60S subunits (both open L1). Heterogeneity analysis was performed with CryoDRGN v.3.0.2 (ref. [55]).

In a second picking round on the same dataset, exclusively particles representing Drg1 alone (not associated with pre-ribosomes) were picked. Initial 2D class averages generated from manually picked particles were used for automated template picking generating an initial set of 3,148,330 particles. Multiple rounds of 2D classification were used to clean the dataset and remove pre-ribosomes. 3D classification by heterogeneous refinement resulted in a final set of 114,728 particles that was used for refinement and 3DVA[56].

3DVA (principal component analysis) allows sorting of structural variants from complex particle mixtures. It was performed on the refined particles in Cryosparc v3.0 with a filter resolution of 7 Å, followed by a 3D Var display job with eight frames. Visualization was performed as a volume series in UCFS ChimeraX v1.3. Conformational heterogeneity in the final particle populations for both the free Drg1 hexamers as well as the Drg1–pre-60S complex was detected and visualized using 3DVA.

After homogeneous refinement in Cryosparc v3.0, particles were transferred to RELION v3.0 using pyEM v0.5 (ref. [57]) for further processing. Finally, maps were post-processed using DeepEMhancer[58].

**Model building and refinement.** As an initial model the symmetric Drg1 hexamer was used (PDB: 7NKU)[17]. Model building was performed in Coot v0.9.6 (ref. [59]), followed by refinement using Phenix v1.18.2-3874 (ref. [60]) and ISOLDE v1.2.2 (ref. [61]).

For the Bud20-TAP particle, initial models of the following components were taken from published early cytoplasmic pre-60S particles: Arx1 was taken from PDB 6RZZ[28], Ribosomal proteins, 25S rRNA, Mrt4, Nog1, Bud20, Rlp24 and YBl028C) were taken from PDB 6N8K[27]. An initial model for L12 was taken from the mature 80S ribosome (PDB 4V6I). As an initial model for the ES27 rRNA segment, PDB 3IZD was used. Adjustment and real-space refinement of these models was performed in Coot v0.9.2 and phenix refine v1.18.2-3874, respectively. For molecular visualization, UCSF chimera v1.14 (refs. [62,63]) and ChimeraX v1.3 were used[64,65].

**Drg1–pre-60S complex crosslinking MS (Arx1-TAP particle).** Pre-60S particles were purified via Arx1-TAP from a thermosensitive drg1-18 mutant after 1 hour of incubation at 37 °C as described previously[12]. After TEV cleavage, the eluate from 2 L of culture was incubated with purified Drg1 (150 μg) in the presence of 1 mM ATPγS for 30 minutes on a rotator at room temperature. Afterwards, one-third of the sample was supplemented with 1.5 mM isotopically labeled crosslinking reagent A (disuccinimidyl suberate, DSS-d0/DSS-d12, Creative molecules) and one-third with 1.5 mM isotopically labeled Crosslinker reagent B (bis(sulfosuccinimidyl)suberate, BS³-d0/BS³-d12, Sigma). One-third was further purified via Calmodulin sepharose beads without crosslinking. Crosslinking reactions were incubated at 30 °C in a thermomixer at 300 r.p.m. After 30 minutes, the reactions were quenched with 5 μl of a 1 M $NH_4HCO_3$ stock solution for further 10 minutes at 30 °C. Subsequently, samples were concentrated in a speedvac and stored at −20 °C.

**Fractionation and enrichment of crosslinked peptides by SEC.** Crosslinked samples were processed essentially as described[28]. In short, samples were dried (Eppendorf, Concentrator plus), resuspended in 8 M urea, reduced, alkylated and digested with trypsin (Promega). Digested peptides were separated from the solution and retained by a solid-phase extraction system (SepPak, Waters). Crosslinked peptides were enriched by size exclusion chromatography (SEC) using an ÄKTAmicro chromatography system (GE Healthcare) equipped with a Superdex Peptide 3.2/30 column (column volume, 2.4 ml). For each crosslinked sample four fractions were measured in technical duplicates. Therefore, the elution fractions 0.9–1.0 and 1.0–1.1 ml were pooled and the three elution fractions 1.1–1.2, 1.2–1.3 and 1.3–1.4 ml were analyzed separately by liquid chromatography with tandem MS (LC–MS/MS). Absorption levels at 215 nm of each fraction were used to normalize peptide amounts prior to LC–MS/MS analysis.

 

**LC–MS/MS analysis.** LC–MS/MS analysis was carried out on an Orbitrap Fusion Tribrid mass spectrometer (Thermo Electron). Peptides were separated on an EASY-nLC 1200 system (Thermo Scientific) at a flow rate of 300 nl/minute over an 80-minute gradient (5% acetonitrile in 0.1% formic acid for 4 minutes, 5%–35% acetonitrile in 0.1% formic acid in 75 minutes, 35%–80% acetonitrile in 1 minute). Full-scan mass spectra were acquired in the Orbitrap at a resolution of 120,000, a scan range of 400–1,500 $m/z$, and a maximum injection time of 50 ms. The most intense precursor ions (intensity $\geq 5.0 \times 10^3$) with charge states 3–8 and monoisotopic peak determination set to 'peptide' were selected for MS/MS fragmentation by CID at 35% collision energy in a data-dependent mode. The duration for dynamic exclusion was set to 60 seconds. MS/MS spectra were analyzed in the ion trap at a rapid scan rate.

**Identification of crosslinked peptides.** MS raw files were converted to centroid files and searched using xQuest/xProphet[66] in ion-tag mode, with a precursor mass tolerance of 10 ppm against a database containing ribosomal proteins and known assembly factors (total of 184 proteins). For each experiment, only unique crosslinks were considered and only high-confidence crosslinked peptides that were identified with a delta score (deltaS) below 0.95 and an ld-Score above 32, translating to an FDR ≤ 0.5, were selected for this study. Crosslink networks were visualized with xiNet[67].

**Expression and purification of GST-Arx1.** *S. cerevisiae* Arx1 was expressed as an N-terminal GST fusion from pGEX-6P-1 in *Escherichia coli* Rosetta (DE3) pLysS cells. Five hundred milliliters of LB medium (+100 µg/ml ampicillin and 40 µg/ml chloramphenicol) were inoculated to a starting $OD_{600}$ 0.04 and grown at 37 °C (170 r.p.m.). At an $OD_{600}$ of 0.4, the culture was shifted to a shaking water bath at 16 °C, and after 30 minutes, heterologous protein expression was induced with 0.4 mM IPTG for 18 hours. Collected cells were washed once with aqua bidest and stored at −80 °C. For cell lysis, cells were resuspended in lysis buffer (50 mM Tris-HCl, pH 7.4, 150 mM NaCl, 1 mM DTT, 0.5 mM PMSF, 1× HP protease inhibitor cocktail (Serva)) and incubated with 1 mg/ml lysozyme for 45 minutes prior to sonication. After removal of cell debris by centrifugation (40,000 g, 25 min, 4 °C), the supernatant was incubated with GSH-agarose beads (Sigma Aldrich) at 4 °C for 60 minutes. Afterwards, beads were washed twice with lysis buffer, once with washing buffer (50 mM Tris-HCl, pH 7.4, 1 M NaCl, 1 mM DTT) and twice with storage buffer (50 mM Tris-HCl, pH 7.4, 150 mM NaCl, 10% vol/vol glycerol, 1 mM DTT). The beads with the bound GST-Arx1 were resuspended in storage buffer, portioned, shock frozen in liquid nitrogen and stored at −80 °C.

**GST pulldown.** GST-Arx1 or GST-Rlp24 beads were thawed and washed once with binding buffer (20 mM HEPES-KOH, 150 mM potassium acetate, 5 mM magnesium acetate, 0.1% Tween-20, 1 mM DTT, pH 6.8). Eighty micrograms Drg1 were incubated with the GST-tagged bait proteins on the beads in binding buffer (120 µl) in the presence of either 1 mM ATP or ATPγS or no nucleotide for 60 minutes at room temperature under constant rotation. Empty GST was included as control for unspecific binding. For the GST-Arx1 pulldown, samples using GST-Rlp24C[12] as bait protein were included as additional reference. After five binding-buffer washing steps, followed by centrifugation (500 g, 1 minute, 4 °C), Arx1 and co-purifying Drg1 were separated from the beads by Prescission protease treatment (GE Healthcare) overnight, which cleaves off the GST tag. Samples using GST or GST-Rlp24C as baits were eluted by addition of 20 mM free GSH (Sigma Aldrich). The eluates were analyzed on a NuPAGE 4–12% Bis-Tris gel (Invitrogen).

**ATPase activity assay (Malachite Green Phosphate Assay).** Drg1 ATPase activity was measured using the Malachite green phosphate assay[68] as reported previously[12,16]. Essentially, purified Drg1 was eluted in 20 mM HEPES-KOH, 150 mM potassium acetate, 5 mM magnesium acetate, 0.1% Tween-20, 1 mM DTT, pH 6.8. HIS$_6$-tagged Rlp24C or the indicated variants thereof were heterologously expressed in *E. coli* and purified as described in ref. [16]. The activity of 5 µg/100 µl Drg1 was measured either alone (basal activity) or in the presence of 2 µg (0.8 µM) HIS$_6$-Rlp24C and/or 2.7 µg (0.4 µM) Arx1. All samples contained 1 mM ATP (Sigma Aldrich). The released phosphate was quantified using the Malachite Green Phosphate Assay kit (Bioassay Systems). The absorbance of the samples at 600 nm was measured at a GeniusPro TECAN plate reader using a Microsoft Excel data collection plugin (XFluor4 v4.51). The specific activity (µmol ATP/hour/mg Drg1) of all samples was normalized to the Drg1 basal activity to display relative activities. Three biological replicates were measured with three technical replicates to determine mean and s.d.

**Spot assays.** Shuffle strains (*drg1Δ*, *drg1Δ/arx1Δ* or *rlp24Δ*) carrying the wild-type genes with their endogenous promoter and terminator on URA3 (pRS316) plasmids were transformed with plasmids expressing the indicated variants of Drg1, Arx1 or Rlp24. For plasmid shuffling, the transformed strains were grown in SD-leu (pRS315 plasmids) or in SD-his-leu (pRS315 and pRS313 plasmids) and spotted in a serial dilution on selective media and 5-FOA agar plates. For overexpression of Drg1 variants from the *CUP1* promoter, the cells were spotted on selective medium containing different concentrations of $CuSO_4$.

**Surface plasmon resonance.** SPR measurements were performed on a Biacore X100 (GE Healthcare/Cytiva). To analyze binding of Drg1 to the Rlp24C domain, purified GST-Rlp24C was immobilized as ligand on a CM5 sensor chip using the amine coupling kit (both Cytiva). Analogously, GST alone was immobilized in the reference flow cell. Drg1 was purified as described and eluted in elution buffer (20 mM HEPES, 10 mM KCl, 2.5 mM $MgCl_2$, 100 mM NaCl, 0.05% Tween-20, 1 mM DTT, pH 7.5), which was also used as running buffer additionally supplemented with 1 mM ATPγS (or 1 mM ATP for the release assay). Then, 75–10,000 nM Drg1 supplemented with 1 mM ATPγS was injected with each cycle, composed of 180 seconds association, 120 seconds dissociation and 60 seconds regeneration of the chip surface with 1 M NaCl. Two biological replicates were measured on two separate CM5 chips with two technical replications ($n = 4$). For binding-affinity determination, measured RUs were normalized to the maximal response of each series (= 100%) and plotted over the concentration in Graphpad prism v3.03. The $K_D$ was determined by non-linear regression (one site binding hyperbola).

To measure binding of Drg1 to the export-competent particles, IgG antibodies were immobilized in both flow cells of CM3 chips (Cytiva) by amine coupling. Subsequently, purified export-competent particles still carrying the Protein A moiety of the TAP tag were captured (120-second contact time, resulting in 1,500–1,700 RU followed by a 30-second stabilization period) on the IgGs prior to injection of increasing concentrations of Drg1. All samples contained 1 mM ATPγS. Each injection cycle was composed of a fresh particle capturing step, followed by injection of Drg1 (180 seconds association, 120 seconds dissociation), and finally two 30-second pulses of 10 mM glycine-HCl pH 2.2 to regenerate the binding surface. Three biological replicates were measured on three separate chips ($n = 3$). For binding-affinity determination, measured RUs were normalized to the maximal response of each series (=100%) and plotted over the concentration in GraphPad prism v3.03. $EC_{50}$ and Hill slope were determined by non-linear regression (sigmoidal dose–response with variable slope).

For the release assay, particles were analogously captured via IgGs, followed by injection of different Drg1 concentrations either containing 1 mM ATPγS (binding control), 1 mM ATP or 1 mM ATP + 100 µg/ml diazaborine. Linear segments (20 seconds) of the 37.5 nM Drg1 injections in the presence of ATP were used to quantify ΔRU/s. To determine the change in the number of bound particles, we used the manufacturer's estimate (GE Healthcare Sensor Surface Handbook) that an SPR signal of 1 RU approximately corresponds to a surface concentration of 1 pg/mm², which is based on an empirical determination using radiolabeled proteins[69] and can further be converted to the volume concentration, taking into account the volume of the dextran matrix (see equation 1 in ref. [70]). Given that the specific responses produced by biomolecules are largely independent of size[69] and that the CM3 sensor chip used for the experiment provides only half the surface volume with the thickness of the dextran matrix being 50 nm, we calculated that a response of ~1,500 RU obtained by the injection of the 2.1 MDa pre-ribosomal particle corresponds to a surface concentration of 0.14 µM. This would give a concentration of ~9.5 nM for 1 RU of the 2.1-MDa pre-ribosomal particle. Finally, we related ΔRU/s to the Drg1 concentration (37.5 nM monomeric Drg1 corresponding to 6.25 nM hexamer) to estimate the rate of particle release per Drg1 hexamer. Four 37.5-nM injections from four biological replicates were used for the quantification ($n = 4$).

**Statistics and reproducibility.** The experiments described in Figures 1a,b and 3c,d,g,h and Extended Data Figs. 3b,d,e and 4 were performed twice, and representative results are shown. All attempts at replication were successful. Reproduction and sample numbers of the experiments described in Figs. 3e,i and 6a,d and Extended Data Fig. 3c are described in the respective figure legends and methods details. No statistical methods were used to predetermine sample size.

**Reporting summary.** Further information on research design is available in the Nature Research Reporting Summary linked to this article.

## Data availability

Cryo-EM maps and coordinate models generated in this study were deposited in the PDB as well as EMDB databases: substrate-bound Drg1 hexamer (PDB: 7Z11, EMDB: EMD-14437) and Drg1-bound to the Bud20-TAP pre-ribosomal particle (PDB: 7Z34, EMDB: EMD-14471). Cryo-EM raw data (unprocessed micrographs) are deposited in the EMPIAR database (accession code EMPIAR-11053). Additional published datasets used in this study are also available from the PDB: for the Bud20-TAP particle, published components of early cytoplasmic pre-60S particles (PDB 6RZZ, 28; (Arx1), PDB 6N8K, 27; (25S rRNA, Mrt4, Nog1, Bud20, Rlp24 and YBl028C) and PDB 6K8K (Nmd3)) were used as initial models. An initial model for L12 was taken from the mature 80S ribosome (PDB 4V6I). As initial model for the ES27 rRNA segment PDB 3IZD was used. The MS raw files, the crosslink database and original xQuest result files have been deposited to the ProteomeXchange Consortium via the PRIDE partner repository[71] with the dataset identifier PXD032098. Source data for the graphs and calculated parameters in Figures 1a,b, 3c–e,g,i and 6a–d and Extended Data Figure 3a–d are provided with this paper as source data files. Source data are provided with this paper.

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

## Acknowledgements

We thank M. Fromont-Racine, A. Johnson, J. Woolford, S. Rospert, J. P. G. Ballesta and E. Hurt for supplying antibodies. The work was supported by Boehringer Ingelheim (to D. H.), the Austrian Science Foundation FWF (grants 32536 and 32977 to H. B.), the UK Medical Research Council (MR/T012412/1 to A. J. W.) and the German Research Foundation (Emmy Noether Programme STE 2517/1-1 and STE 2517/5-1 to F.S.). We thank Norberto Escudero-Urquijo, Pablo Castro-Hartmann and K. Dent, Cambridge Institute for Medical Research, for their help in cryo-EM during early phases of this project. This research was supported by the Scientific Service Units of IST Austria through resources provided by the Electron Microscopy Facility. We thank S. Keller, Institute of Molecular Biosciences (Biophysics), University Graz for support with the quantification of the SPR particle release assay. We thank I. Schaffner, University of Natural Resources and Life Sciences, Vienna for her help in early stages of the SPR experiments.

## Author contributions

M. P., I. G., D. H. and H. B. designed the study and planned the experiments. C. H. and G. Z. purified the proteins and the pre-ribosomal particles. V.-V. H. froze the samples and collected the cryo-EM data. V. K. and A. J. W. generated initial cryo-EM maps of Drg1–pre-60S complexes. I. G., M. P., D. H. and H. B. processed the data and generated the maps. I. G., M. P., H. B. and D. H. built the models. G. Z. performed the crosslinking reaction, and C. S. and F. S. performed the MS analysis. C. H. and M. P. performed the SPR analysis. C. H. performed the Arx1 pulldown and the Drg1 ATPase activity assay. M. L., M. G. and G. Z. generated constructs and performed dot spot assays. L. K. and M. G. performed the fluorescence microscopy. M. P., I. G., D. H. and H. B. generated the figures. M. P., I. G., D. H. and H. B. wrote the manuscript. All authors read and approved the final version of the manuscript.

## Competing interests

The authors declare no competing interests.

## Additional information

**Extended data** is available for this paper at https://doi.org/10.1038/s41594-022-00832-5.

**Correspondence and requests for materials** should be addressed to David Haselbach or Helmut Bergler.

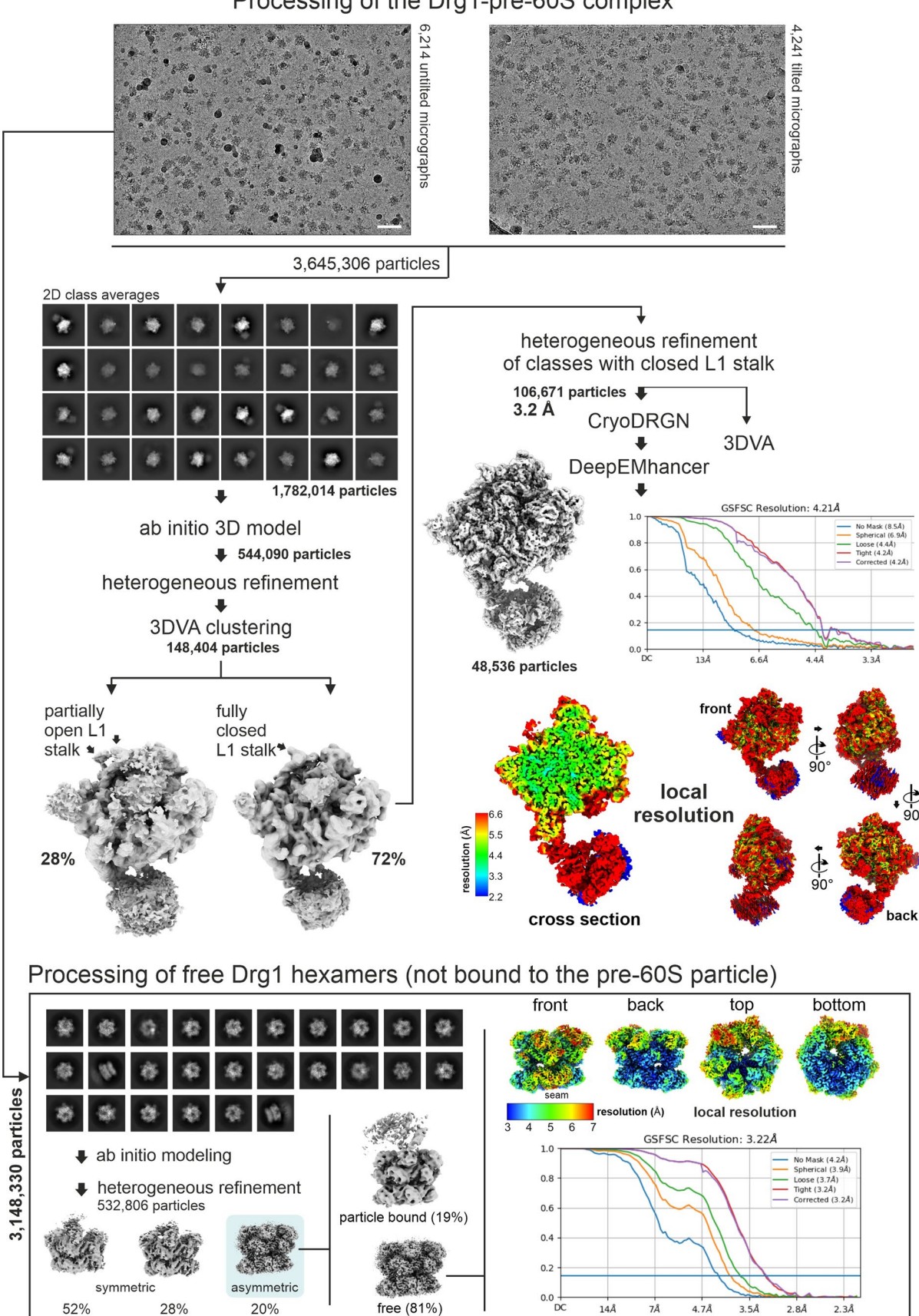

**Extended Data Fig. 1 | Cryo-EM data processing scheme.** Processing of the Drg1-pre-60S complex and free Drg1 hexamers showing preprocessed micrographs (scale bar = 500 Å), 2D class averages, 3D classification and refinement steps as well as local resolution estimation.

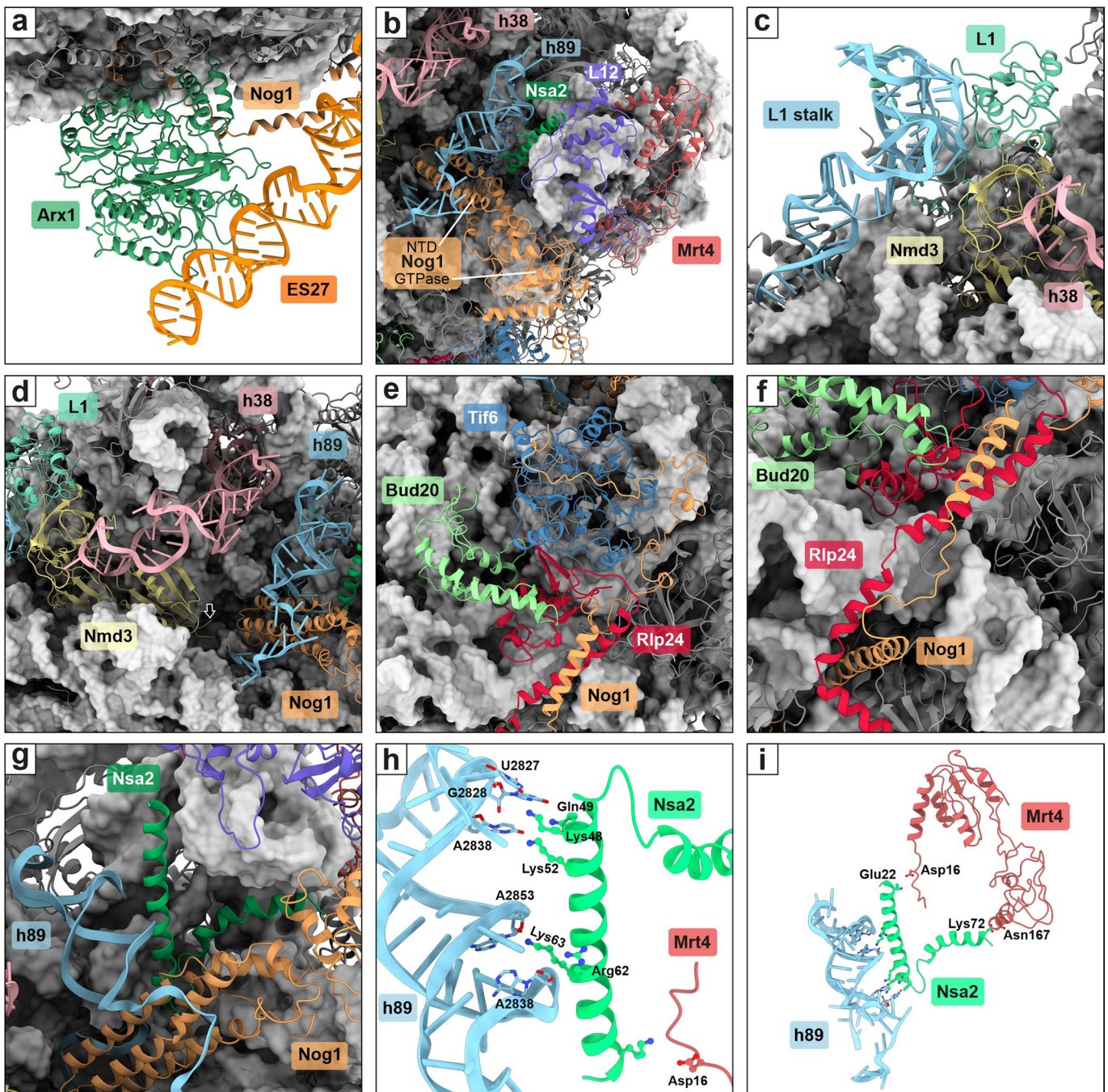

**Extended Data Fig. 2 | Structural details of the export-competent Bud20-TAP pre-60S particle (rRNAs that are not highlighted are displayed as grey surfaces). (a)** Drg1 docking platform composed of Arx1 and ES27. **(b)** Localization of the ribosomal protein Rpl12 (L12) and the shuttling factors Nog1, Nsa2, Mrt4, Nmd3, Nog1 as well as 25 S rRNA helices 38 (h38) and 89 (h89). **(c)** Closed L1 stalk. Localization of ribosomal protein L1. **(d)** The N-terminus of Nmd3 is not resolved in the structure. The black arrow marks the first resolved residue of Nmd3 (R156). **(e)** and **(f)** Entanglement of Bud20, Rlp24, Nog1 and Tif6. **(g)** Localization of the two N-terminal Nsa2 helices (His20 to Lys74) between h89 and Mrt4. **(h)** Interaction surface between Nsa2 and 25 S rRNA helix 89. **(i)** Nsa2 connects Mrt4 and helix 89.

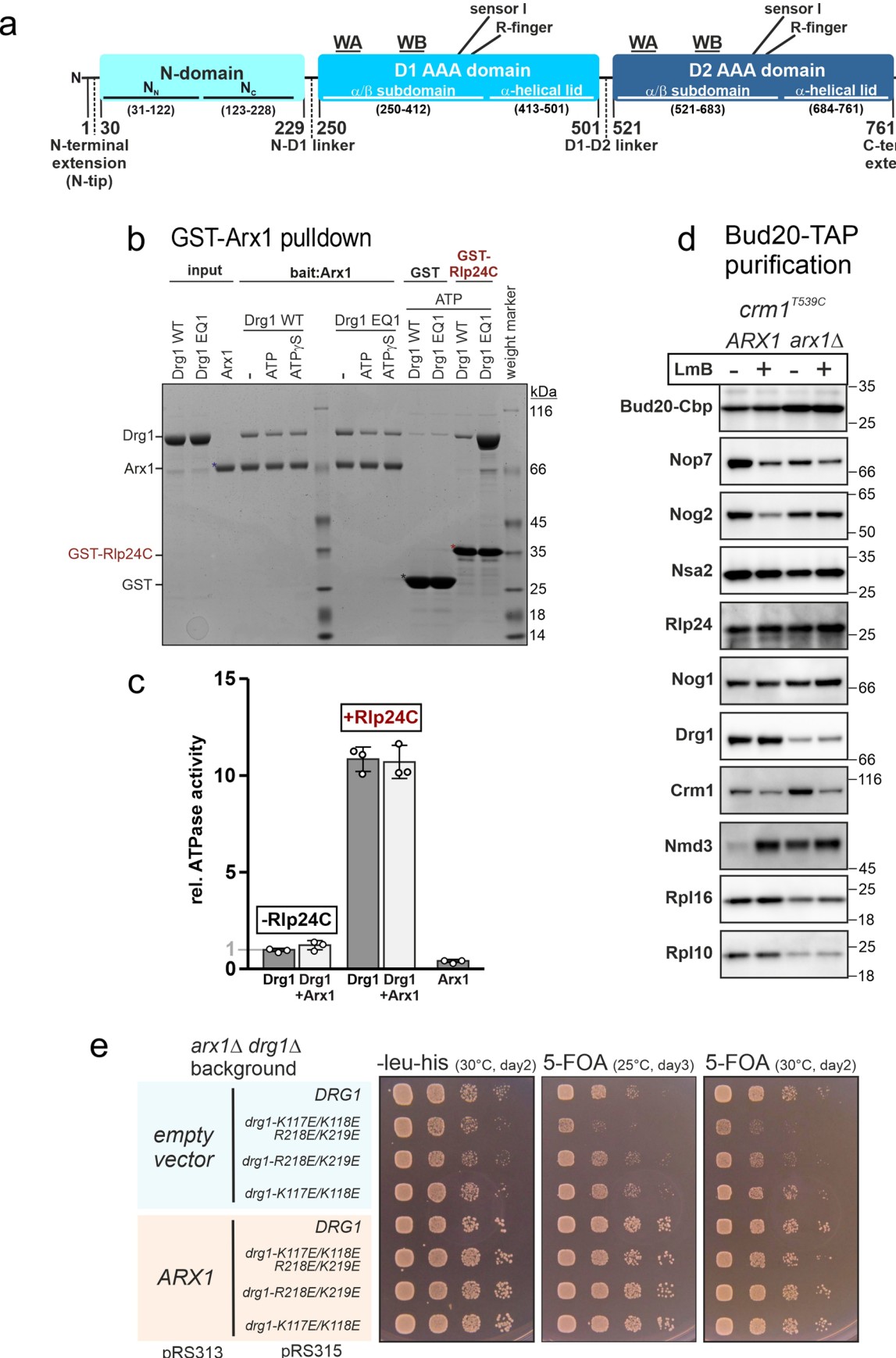

**a**

**b** GST-Arx1 pulldown

**c**

**d** Bud20-TAP purification

**e**

Extended Data Fig. 3 | See next page for caption.

**Extended Data Fig. 3 | Role of Arx1 in the recruitment of Drg1 to the pre-60S particle. (a)** Domain organization of Drg1. **(b)** GST-Arx1 pulldown using Drg1 wildtype and the D1 Walker B mutant Drg1EQ1 (E346Q) as prey in the presence of 1 mM ATP, ATPγS or no nucleotide (-). GST served as control for unspecific binding, GST-Rlp24C as binding reference. **(c)** *In vitro* ATPase activity assay of Drg1 in the presence of purified Arx1. Relative activity normalized to Drg1 wildtype basal activity. Rlp24C = HIS$_6$-tagged C-terminal 53 amino acids of Rlp24. Mean ± standard deviation of three biological replicates ($n = 3$). **(d)** Composition of Bud20-TAP particles from the Leptomycin B (LmB) sensitive crm1$^{T539C}$ mutant strain with either *ARX1* wildtype or *Δarx1* background (Western blot). **(e)** Spot assay. For plasmid shuffling, a *Δdrg1 Δarx1* double shuffle strain carrying wildtype *DRG1* on a URA3 plasmid (pRS316) as well as either pRS313-*ARX1* or the empty vector control (pRS313) was transformed with the indicated plasmids carrying mutant *drg1* alleles and spotted on selective media and 5-FOA agar plates.

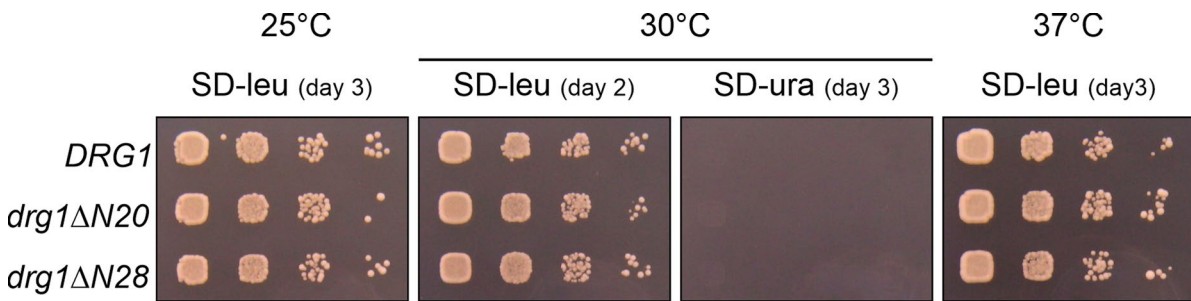

**Extended Data Fig. 4 | Functionality of N-terminal Drg1 truncations.** *drg1Δ* shuffle strains expressing N-terminally truncated Drg1 variants were spotted on SD-Leu, SD-Ura and 5-FOA agar plates after plasmid shuffling on 5-FOA.

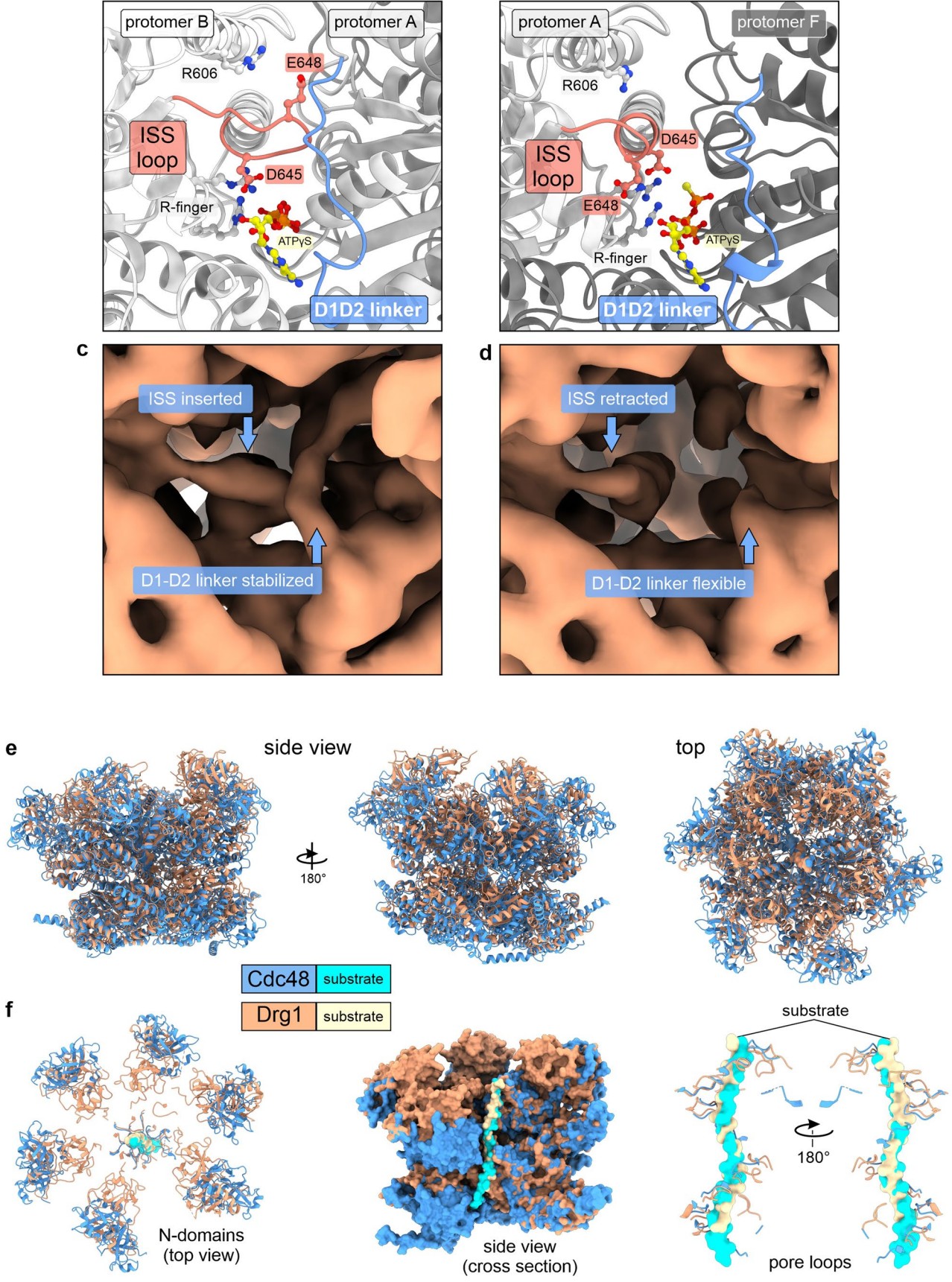

**Extended Data Fig. 5 | See next page for caption.**

**Extended Data Fig. 5 | Positioning of the ISS. (a)** The ISS loop of Drg1 protomer B is fully inserted into the nucleotide binding pocket of the adjacent protomer A. **(b)** The ISS loop of protomer A is retracted from the nucleotide binding pocket of the adjacent protomer F. **(c)** and **(d)** Interaction with the inserted ISS loop stabilizes the D1D2 linker **(c)**, while it is flexible and not well-defined in the cryo-EM map when the ISS loop is retracted **(d)**. **(e)** and **(f)** Comparison of substrate translocation between Drg1 and Cdc48 (PDB: 6ONC) with a superposition of the two hexamers **(e)** and highlighted N-domain positioning relative to the inserted substrate as well as engagement of the substrate polypeptide chain inside the central channel via the pore loops **(f)**.

# nature research

# Reporting Summary

Nature Research wishes to improve the reproducibility of the work that we publish. This form provides structure for consistency and transparency in reporting. For further information on Nature Research policies, see our Editorial Policies and the Editorial Policy Checklist.

## Statistics

For all statistical analyses, confirm that the following items are present in the figure legend, table legend, main text, or Methods section.

| n/a | Confirmed | |
|---|---|---|
| ☐ | ☒ | The exact sample size (*n*) for each experimental group/condition, given as a discrete number and unit of measurement |
| ☐ | ☒ | A statement on whether measurements were taken from distinct samples or whether the same sample was measured repeatedly |
| ☒ | ☐ | The statistical test(s) used AND whether they are one- or two-sided *Only common tests should be described solely by name; describe more complex techniques in the Methods section.* |
| ☒ | ☐ | A description of all covariates tested |
| ☒ | ☐ | A description of any assumptions or corrections, such as tests of normality and adjustment for multiple comparisons |
| ☐ | ☒ | A full description of the statistical parameters including central tendency (e.g. means) or other basic estimates (e.g. regression coefficient) AND variation (e.g. standard deviation) or associated estimates of uncertainty (e.g. confidence intervals) |
| ☒ | ☐ | For null hypothesis testing, the test statistic (e.g. *F*, *t*, *r*) with confidence intervals, effect sizes, degrees of freedom and *P* value noted *Give P values as exact values whenever suitable.* |
| ☒ | ☐ | For Bayesian analysis, information on the choice of priors and Markov chain Monte Carlo settings |
| ☒ | ☐ | For hierarchical and complex designs, identification of the appropriate level for tests and full reporting of outcomes |
| ☒ | ☐ | Estimates of effect sizes (e.g. Cohen's *d*, Pearson's *r*), indicating how they were calculated |

*Our web collection on statistics for biologists contains articles on many of the points above.*

## Software and code

Policy information about availability of computer code

| Data collection | Raw data (OD600) of the ATPase activity measurements were collected on a Tecan plate reader using an associated Microsoft excel macro plugin (XFluor4 v4.51). Cryo-EM data were recorded with SerialEM v3.8 (https://bio3d.colorado.edu/SerialEM/). SPR data were collected using the Biacore X100 control software v2.0.2 (Cytiva). Western blots and SDS gels were imaged on a ChemiDoc Touch imaging system (Biorad). |
|---|---|
| Data analysis | Analysis of biochemical/biophysical data (ATPase activity and SPR measurements): Microsoft Excel 2019 and the Graphpad Prism software v 3.03, Biacore X100 evaluation software (SPR). Western blot analysis: ImageLab software v.2.2.0.08 (BioRad). Crosslinking MS: xiNET v1.1.13 (Rappsilber Laboratory) and xQUest/xProphet v2.1.5 (https://bioinformaticshome.com). Cryo-EM structure analysis: Coot v0.9.2/v0.9.6, CryoDRGN v0.3.2 (https://github.com/zhonge/cryodrgn), Cryosparc v3.0, DeepEMhancer (https://github.com/rsanchezgarc/deepEMhancer), Isolde v.1.2.2 (https://github.com/tristanic/isolde), PHENIX suite v1.18.2-3874, RELION v3.0, Rosetta v3.0, UCSF Chimera v.1.15, UCSF ChimeraX v1.1.1, UCSF pyem v0.5. |

For manuscripts utilizing custom algorithms or software that are central to the research but not yet described in published literature, software must be made available to editors and reviewers. We strongly encourage code deposition in a community repository (e.g. GitHub). See the Nature Research guidelines for submitting code & software for further information.

## Data

Policy information about availability of data

All manuscripts must include a data availability statement. This statement should provide the following information, where applicable:

- Accession codes, unique identifiers, or web links for publicly available datasets
- A list of figures that have associated raw data
- A description of any restrictions on data availability

Cryo-EM maps and coordinate models generated in this study were deposited in the PDB as well as EMDB databases: Substrate-bound Drg1 hexamer (PDB: 7Z11,

EMDB: EMD-14437) and Drg1-bound to the Bud20-TAP pre-ribosomal particle (PDB: 7Z34, EMDB: EMD-14471). Cryo-EM raw data (unprocessed micrographs) are deposited in the EMPIAR database (accession code EMPIAR-11053). Additional published datasets used in this study are available from the PDB: For the Bud20-TAP particle, published components of early cytoplasmic pre-60S particles (PDB 6RZZ, 28; (Arx1), PDB 6N8K, 27; (25S rRNA, Mrt4, Nog1, Bud20, Rlp24 and YBl028C) and PDB 6K8K (Nmd3) were used as initial models. An initial model for L12 was taken from the mature 80S ribosome (PDB 4V6I). As initial model for the ES27 rRNA segment PDB 3IZD was used. The MS raw files, the crosslink database and original xQuest result files have been deposited to the ProteomeXchange Consortium via the PRIDE partner repository with the dataset identifier PXD032098. Source data for the graphs and calculated parameters in figures 1a, 1b, 3c-e and 3g-i, 6a-d and extended data figures 3a-d are provided with this paper as Source data files.

# Field-specific reporting

Please select the one below that is the best fit for your research. If you are not sure, read the appropriate sections before making your selection.

☒ Life sciences ☐ Behavioural & social sciences ☐ Ecological, evolutionary & environmental sciences

For a reference copy of the document with all sections, see nature.com/documents/nr-reporting-summary-flat.pdf

# Life sciences study design

All studies must disclose on these points even when the disclosure is negative.

| | |
|---|---|
| Sample size | Sample sizes (n) are supplied in the figure legends as well as the methods details. No mathematical sample size calculation was performed. All biochemical and yeast growth experiments were performed with multiple biological and technical replicates to allow estimation of the distribution of the data. Sample sizes are based on preliminary and published studies (Kappel et al., 2012, Loibl et al., 2014, Prattes et al., 2017, 2021) and were determined by number of replicates necessary to ensure reproducibility. Detailed information for the individual experiments including sample size and replicates are stated in the figure legends, the methods section as well as the source data file provided along with this paper. |
| Data exclusions | Individual cryo-EM micrographs were discarded due to strong drift, devitrification or ice contamination after manual inspection. For the biochemical measurements and yeast growth experiments no datasets were excluded. |
| Replication | For all biochemical measurements, 2-4 biological replicates were tested, each measured with at least two technical replications. All attempts at replication were successfull. Detailed information for the individual experiments, including exact sample number (n) are stated in the figure legends, the methods section as well as the source data file provided along with this paper. |
| Randomization | For calculation of the Fourier Shell Correlation using Cryosparc, the cryo-EM particles were automatically split into two random halves by the software. For biochemical/biophysical analyses, samples from each biological replicate were randomly assigned to the tested conditions. For growth comparison experiments of different yeast strains, randomization was not applicable. |
| Blinding | For biophysical/biochemical and yeast growth experiments, the same investigator performed data collection and/or analysis. Data acquisition and analyses were performed using constant parameters across all conditions being tested. The quantitative readout of the biochemical/biophysical assays in this study did not require subjective interpretation of the results and thus blinding was not applied. |

# Reporting for specific materials, systems and methods

We require information from authors about some types of materials, experimental systems and methods used in many studies. Here, indicate whether each material, system or method listed is relevant to your study. If you are not sure if a list item applies to your research, read the appropriate section before selecting a response.

## Materials & experimental systems

| n/a | Involved in the study |
|---|---|
| ☐ | ☒ Antibodies |
| ☒ | ☐ Eukaryotic cell lines |
| ☒ | ☐ Palaeontology and archaeology |
| ☒ | ☐ Animals and other organisms |
| ☒ | ☐ Human research participants |
| ☒ | ☐ Clinical data |
| ☒ | ☐ Dual use research of concern |

## Methods

| n/a | Involved in the study |
|---|---|
| ☒ | ☐ ChIP-seq |
| ☒ | ☐ Flow cytometry |
| ☒ | ☐ MRI-based neuroimaging |

## Antibodies

| | |
|---|---|
| Antibodies used | Rabbit α-Cbp: Sigma - Aldrich Cat# SAB4500455; Rabbit α-Crm1 (1:10,000): C. Yan (Yan et al., 1998); Rabbit α-Drg1 (1:5,000): Zakalskiy et al., 2002; Rabbit α-Mex67 (1:10,000): E. Hurt/Segref et al., 1997; Rabbit α-Mrt4 (1:1,000): Jesus de la Cruz (Rodriguez-Mateos et al., 2009); Rabbit α-Nmd3 (1:4,000): A. W. Johnson (Kallstrom et al., 2003); Rabbit α-Nog1 (1:5,000): M. Fromont-Racine (Saveanu et al., 2003); Rabbit α-Nog2(1:5,000): M. Fromont-Racine (Saveanu et al., 2001); Rabbit combined α-Ytm1 and α-Nop7(1:5,000): J. d. l. Cruz (Wegrecki et al., 2015); Rabbit α-Nsa2: (1:5,000) M. Fromont-Racine (Lebreton et al., 2006); Rabbit α-Rlp24 (1.5,000): M. Fromont-Racine (Saveanu et al., 2001); Rabbit α-Rpl10 (1:10,000): B. L. Trumpower (Eisinger et al., 1997); Rabbit |

α-Rpl16 (1:40,000): S. Rospert (Peisker et al., 2008); Rabbit α-Rsa4 (1:10,000): M. Remacha (de la Cruz et al., 2005); Peroxidase conjugated Goat α-Rabbit IgG Antibody (sec. AB; 1:10,000): Sigma - Aldrich Cat# A0545

Validation

Commercially available antibodies were validated by the manufacturer. All antibodies are from published studies and were used previously (Pertschy et al., 2007, Kappel et al., 2012, Bassler et al., 2012, Zisser et al., 2018).

