## [Peer Review File · Nature Structural & Molecular Biology]

Peer Review Information

Journal: Nature Structural and Molecular Biology

Manuscript Title: Visualizing maturation factor extraction from the nascent ribosome by the AAA-ATPase Drg1

Corresponding author name(s): Professor Helmut Bergler

Editorial Notes:

Reviewer Comments & Decisions:

Decision Letter, initial version:

31st Jan 2022

Dear Dr. Bergler,

Thank you again for submitting your manuscript "Visualizing maturation factor extraction from the nascent ribosome by the AAA-ATPase Drg1". I apologize for the delay in responding, which resulted from the difficulty in obtaining suitable referee reports. Nevertheless, we now have comments (below) from the 3 reviewers who evaluated your paper. In light of those reports, we remain interested in your study and would like to see your response to the comments of the referees, in the form of a revised manuscript.

You will see that some concerns were raised about confirmation of the findings by biochemical assays and mutagenesis, as well as specific questions regarding the variability analysis and discussion of the AAA+ mechanism. Editorially, we agree that these suggestions would strengthen the work, and ask that they be incorporated in a revised manuscript. Please be sure to address/respond to all concerns of the referees in full in a point-by-point response and highlight all changes in the revised manuscript text file. If you have comments that are intended for editors only, please include those in a separate cover letter.

We expect to see your revised manuscript within 6 weeks. If you cannot send it within this time, please contact us to discuss an extension; we would still consider your revision, provided that no similar work has been accepted for publication at NSMB or published elsewhere.

Reporting Summary:

When submitting the revised version of your manuscript, please pay close attention to our [href="https://www.nature.com/nature-research/editorial-policies/image-integrity">Digital Image Integrity Guidelines. and to the following points below:](https://www.nature.com/nature-research/editorial-policies/image-integrity)

[FOR STRUCTURAL MS] If there are additional or modified structures presented in the final revision, please submit the corresponding PDB validation reports.

[FOR MS WITH CROPPED GELS] Please note that all key data shown in the main figures as cropped gels or blots should be presented in uncropped form, with molecular weight markers. These data can be aggregated into a single supplementary figure item. While these data can be displayed in a relatively informal style, they must refer back to the relevant figures. These data should be submitted with the final revision, as source data, prior to acceptance, but you may want to start putting it together at this point.

Data availability: this journal strongly supports public availability of data. All data used in accepted papers should be available via a public data repository, or alternatively, as Supplementary Information. If data can only be shared on request, please explain why in your Data Availability Statement, and also in the correspondence with your editor. Please note that for some data types, deposition in a public repository is mandatory - more information on our data deposition policies and available repositories can be found below:
<https://www.nature.com/nature-research/editorial-policies/reporting-standards#availability-of-data>

[Redacted]

Sincerely,
Sara

Sara Osman, Ph.D.
Associate Editor
Nature Structural & Molecular Biology

Referee expertise:

Referee #1: Functional analysis, ribosome biogenesis

Referee #2: Structural and biophysical analysis, ribosome biogenesis

Referee #3: CryoEM

Reviewers' Comments:

Reviewer #1:

Remarks to the Author:

Prattes et al. (NSMB-A45578) report cryo-EM structures of an export-competent pre-60S particle, purified via the Bud20-TAP bait from cells treated with the export inhibitor leptomycin B, that either contain or lack exogenously added Drg1, a hexameric type II AAA-ATPase that was previously shown by the Bergler laboratory to mediate the release of the biogenesis factor Rlp24 from cytoplasmic pre-60S particles. The visualized structures of Drg1-bound pre-60S particles, together with the high-resolution structure of free Drg1 in the substrate-processing state, reveal how: 1) Drg1 is recruited, via the N-terminal domain of one of the six subunits, to its pre-60S docking site, which is mainly composed of Arx1 and rRNA expansion segment ES27; 2) the tips of four N-terminal domains contact the Rlp24 substrate and guide its C-terminal residues to the central pore of the hexameric ring-like structure; 3) Drg1 mechanistically exerts substrate threading through its central pore.

While the revealed substrate threading mechanism of Drg1 is similar to the one of the related AAA-ATPase Cdc48/p97, which is recruited to and segregates a broad range of ubiquitinated substrate proteins from membranes and stable complexes, it is novel with respect to how such an AAA-ATPase specifically associates with a macromolecular complex in order to be ideally positioned to extract its exclusive substrate protein. Notably, this study not only reveals how Drg1 achieves extraction of Rlp24 (and the neighboring Bud20) from cytoplasmic pre-60S subunits, but also indicates how this enforces larger conformational changes that prime the pre-60S subunit for downstream maturation events.

Taken together, this study provides for the first time molecular insights into the mechanism of Drg1

action and highlights how this AAA-ATPase initiates the cascade of cytoplasmic pre-60S maturation events. These findings are not only exciting for the ribosome biogenesis field but are also of importance for our general understanding of the modus operandi of type II AAA-ATPases.

Comments and suggestions:

- 1) Figure 2c: Based on the cryo-EM structure, one of the N-terminal domains of Drg1, by interacting with Arx1 and ES27, anchors the AAA-ATPase at the pre-60S particle in proximity of Rlp24. To investigate the importance of these interactions for recruitment of Drg1, it could be tested whether absence of Arx1 in combination with mutation of Drg1's ES27-interacting residues (K146, K150, K219) results in a synergistic growth defect.
- 2) Results, line 284 and legend to Figure 3c: Based on the pronounced dominant negative growth phenotype elicited by ectopic expression of a Drg1 protein lacking the first 28 residues, the authors conclude that the "N-terminally truncated Drg1 variants are nonfunctional" (line 284) and that "this part of the protein is essential" (legend Figure 3c, line 267/268). In order to unambiguously show that the tip region of the N-terminal domain indeed fulfils an essential role for substrate recognition and guiding the C-terminal residues of Rlp24 into the D1 ring, the ability of the Drg1 Δ N20 and Drg1 Δ N28 variants to complement the absence of Drg1 should be assessed (as done in Figure 3e for the C-terminally truncated Rpl24 proteins).
- 3) Results, line 288-290: The authors state that "Thus, the very C-terminal end of Rlp24 is crucial for interaction and stimulation of Drg1." and then conclude that "Due to the specificity of Drg1 for Rlp24, the recruitment has to be sequence specific.". First, it has been previously shown that Rlp24 Δ C, lacking the complete C-terminal domain, still interacts with Drg1 (Kappel et al. J. Cell Biol. 2012). Second, assuming that the human Drg1 ortholog SPATA5 would also mediate extraction of the Rlp24 ortholog in a similar manner, it is surprising that the C-terminal domain/tail is not conserved between the yeast and human Rlp24 protein. Further, how do the authors reconcile the proposed sequence-specific recognition and threading of Rlp24's C-terminal tail with the observation that strains expressing C-terminally TAP-tagged Rlp24 have been reported to be viable (Saveanu et al. Mol. Cell. Biol. 2003, Pertschy et al. Mol. Cell. Biol. 2007). Can Drg1, like Cdc48 (ubiquitin), also process stably-folded substrates? To confirm that Rlp24-TAP is indeed (partially) functional, I encourage the authors to assess complementation of the Δ rlp24 shuffle strain by Rlp24-TAP expressed from plasmid. At the same time, it would also be informative to test C-terminally GFP-tagged Rlp24 for complementation.
- 4) Crosslinking MS (Supplementary Table S2): How do the authors explain that crosslinks between Drg1 and early biogenesis factors (Drs1, Erb1, Nop8, etc.), which are not expected to be present on the same pre-ribosomal particles as Drg1, are detected in these experiments? Does this question the specificity of the reported crosslinking data?
- 5) Figure 1a: Why is Drg1 present in good amounts on Bud20-TAP pre-60S particles isolated from leptomycin B-treated cells? Should the isolated particles not correspond to nuclear pre-60S particles whose export to the cytoplasm is inhibited and should therefore not contain Drg1? Can the authors provide an explanation for this observation?
- 6) Discussion, line 540-542: The authors mention only briefly once that loosening of Rlp24 from the complex may involve an entropic pulling force generated by the movement of Drg1 relative to the particle. Can the authors provide references to articles here that highlight how ATPases utilize entropic pulling forces to act on substrates? On the other hand, the phrasing of many other sentences suggests that Drg1 uses mechanical force or acts similarly as a 'Brownian Ratchet' to extract Rlp24: "The coordinated consecutive grabbing and releasing of the substrate by the pore loops creates directionality and stepwise pulls the substrate chain through the pore. (line 447-449)", "The unphased behavior of the D1 and D2 rings likely guarantees a constant unidirectional force generation required to pull off Rlp24 from the pre-ribosome and thus counteract the peptide's chain backtracking. (line

574-576)" and "Taken together, ... and dissect the chain of events during mechanical extraction of its specific substrate. (line 602-604)". I feel that it might help to more clearly discuss the different possibilities, maybe they are not even mutually exclusive.

7) Discussion, line 566-568: The authors mention only briefly that Drg1-mediated extraction of Rlp24 "... follows a conserved scheme in type II AAA-ATPases.". If possible, can they provide a Supplementary Figure comparing (or overlaying) the substrate-processing Drg1 and Cdc48 structures to visualize the similarities?

Minor points:

- 1) Abstract, line 38 (and also later on): eukaryote-specific instead of 'eukaryotic specific'
- 2) Introduction, line 58: remove the definite article before 5.8S.
- 3) Introduction, line 65 (and also later on): Crm1/Xpo1 instead of 'Crm1/XpoI'
- 4) Results, line 94-96: It might help to also state in the text (and not only in the legend to Figure 1a) that the visualized pre-60S particles were purified via Bud20-TAP. I would therefore suggest to rephrase the sentence as follows: "We assembled pre-60S particles, purified via the Bud20-TAP bait, with wild-type Drg1"
- 5) Results, line 145: Mrt4 instead of 'Mtr4'
- 6) Results, line 504-506: This sentence seems to be incomplete. It could be changed to: 'Taken together, our results show that Drg1 is highly efficient in releasing Rlp24 (and Bud20) from nascent pre-60S particles and that the release strictly depends on ATP hydrolysis in the D2 domain.'
- 7) Two of the used strains are not listed in the strain table: Bud20-GFP crm1.T539C (used in Figure 1b) and Arx1-TAP drg1-18 (used for crosslinking MS).

Reviewer #2:

Remarks to the Author:

The manuscript "Visualizing maturation factor extraction from the nascent ribosome by the AAA-ATPase Drg1", by Prattes et al. provides a structural exploration into how Drg1 removes Rpl24 from pre-60S particles in order to drive ribosome maturation. Their data identify important domains and regions for substrate recognition and processing, and confirm the stereotypic AAA "hand-over-hand" model. Overall, the completed work is very impressive, and I believe it will be a useful addition to the field and is appropriate for publication in NSMB, after my concerns are addressed.

Concerns/comments:

1. Figure 1a: Why is there signal for Drg1 in the Western Blot for LmB-treated cells? Are these gels run after overexpressed Drg1 was added? If not, the authors should discuss. Further towards that end, the main text (section starting on line 92) should more explicitly say that you are adding separately-expressed and purified Drg1 to the native particles prior to freezing.
2. Line 175: This is too strong of an interpretation, and should either be moved to the discussion or removed. Most notably, the structure is in the presence of ATPγS, not ATP (and reasonably so), where force is not being generated by the motor. The ring may very well be more "plush" against the pre-60S in ATP – we don't know. Using only the ATPγS structure, these claims are far too speculative for the Results section.
3. Fig 2D and the associated Results section (line 235) are opaque to me. It is difficult to fully assess if this variability analysis is sufficient to justify the strong claims on specific changes to protein-protein interactions. A more detailed explanation of the 3D variability analysis protocol used should be provided, and the limitations of this analysis should be discussed.
4. Line 282 and 334: The claim that four N-domains engage the substrate is too strong to be justified

by the data. Engagements are only qualified to within the limits of the resolution of the structure. Only one nucleotide state – ATP γ S – is assessed. A different number of N-domains may engage in different ATP/ADP states. These limitations must be explicitly discussed, or the claim should be dropped.

5. I am concerned that the claim “The Drg1 N-domain tips coordinate initial substrate recognition” is not fully justified by the data (Fig. 3 title, line 280, line 294, Fig. 5a). How do you know that the N-domain forms contacts prior the D1D2 domains temporally? There is no data to support this.

Engagement is only showed in a single nucleotide state. It is perfectly possible that the N-domains don't engage with the substrate until after it is already in the pore, and serve some important role (e.g., preventing slippage).

6. Figure 5 and Results Section “Substrate extraction via hand-over-hand translocation: It is not clear to this reviewer how information about nucleotide-dependent structural changes can be extracted from a 3D variability analysis of a single (ATP γ S) structure.

7. Fig. 6C: The Rlp24C binding data does not plateau. In such a case, it is improper to normalize to the higher data point. A fitted EC50 and Hill sloped can not be determined. All that can be determined in a lower bound for the EC50, i.e. the EC50 is $> 1 \mu\text{M}$. The Bud20 data barely plateaus. The authors should repeat this experiment at high Drg1 concentrations to ensure a stable upper plateau of binding.

8. Fig 6D: The release rate estimate depends strongly on the rough estimate of 1000 RU equaling 3 mg/mL. I am concerned that this estimate is inaccurate. Deposition of the particles leads to a response of 1,500-1,700 RU, which should accordingly be $\sim 4.5 \text{ mg/mL}$. What is the particle concentration in mg/mL? Is it reasonable to be injecting/depositing 4.5 mg/mL? Ideally, the authors will either devise a more quantitative way of converting RU to nM, or explicitly state the limitations and assumptions built into their measurements, and appropriately pare back language to match the uncertainty. Furthermore, if deposition leads to 1,500 RU, why does addition of 75 nM Drg1+ATP lead to only ~ 400 RU at the plateau? Presumably particles cannot rebind after being processed by Drg1.

Minor Comments

1. Line 61: Missing parenthesis

2. Line 92: Please be more explicit in stating the model organism, strain, and pre-60S isolation strategy in the main text.

3. Line 137: The authors should discuss here the average resolution of the CryoEM structure. Is it sufficient to build in side chains, etc.

4. Figure 1a-b legend: Please define the “0” condition (first lane in gel).

5. Line 167: Please provide a quantitation of the “minor population” (e.g., relative # of particles). Separately, it would be helpful if the authors could add color to the important elements in the “additional contact site” box, similar to how they did for the “docking platform” box

6. Fig 2a: A linear domain diagram of Drg1 would be helpful, including aa cutoffs for the Nn and Nc subdomains.

7. Fig S3C: Please state in the caption what the data points represent (technical or biological replicates), as well as what the error bars represent (presumably mean \pm SD)

8. Line 202-203: What data support this claim? This statement should be in the Discussion not Results, with an appropriate qualifier

9. Line 214: Please perform mutagenesis analysis to specifically and definitely prove that RNA contacts to the N-domain are essential, or remove this claim of primacy.

10. Fig 3a caption: Please state the threshold used

11. Line 290: “has to be sequence specific” is uncomfortably strong language. Rlp24 is currently the known substrate of Drg1, but others may exist and be uncharacterized.

12. Line 361: Again, please acknowledge in the text that this interpretation is only made based on ATP γ S data. Further work is needed to know if the D1 and D2 domains are significantly out of phase in

the

Reviewer #3:

Remarks to the Author:

The manuscript entitled "Visualizing maturation factor extraction from the nascent ribosome by the AAA-ATPase Drg1" reports a series of cryoEM structures of the maturing large ribosomal subunit in complex with the ribosome assembly factor Drg1. Drg1 is a type II AAA+ ATPase and catalyses the first pre60S particle maturation step after nuclear export, the removal of Rpl24. The authors use a chemical inhibitor to enrich Drg1 depleted pre60S particles and reconstitute the complex by adding recombinantly purified Drg1. The functionality of this in-vitro reconstituted complex - the ability to extract Rpl24 in the presence of ATP - is demonstrated by SPR release assays.

The authors visualize pre60S remodelling events caused by the binding of the Drg1 hexamer and are able to analyse the Drg1 recruitment interface formed by Arx1, ES27 rRNA and helix 98 of the 25S-RNA. The presented cryoEM structures of the complex also reveal the initial steps of Rpl24 capture by the Drg1 N-terminal domain. Using high-resolution cryoEM structures of free, substrate engaged Drg1 hexamers, they can analyse the mechanism of Rpl24 unfolding by the Drg1 hexamer. Similar to other AAA+ unfoldases, Rpl24 unfolding proceeds via a classical "hand-over-hand" mechanism, which involves loop regions harbouring conserved tyrosine residues that thread Rpl24 through the central pores of the AAA+ rings. The authors are also able to indirectly quantify the Drg1 mediated Rpl24 release rate from pre60S particles using a SPR approach.

All experiments seem solid and the presented conclusions are justified. The cryoEM maps have been carefully analysed and not over interpreted. The structural hypotheses developed based on the presented cryoEM maps are nicely supported by additional biochemical experiments: The Drg1 - pre60S recruitment interface is validated by crosslinking mass spectrometry, the nucleotide independency of the Arx1 - Drg1 interaction by pull down assays and the importance of the interaction between the C-terminus of Rpl24 and the N-terminal Drg1 domain for initial engagement by yeast growth assays and ATPase assays.

The manuscript highlights a key step in the maturation of the large ribosomal subunit at unprecedented detail for the first time. The pre60S-Drg1 and Drg1 structures provide novel insights into Drg1 recruitment to pre60S particles and the capture of its Rpl24 substrate. The presented results are an important contribution to ribosome maturation field and will be of great interest for the AAA+ community as well. I recommend publication in NSMB.

Minor points:

- The presented structures were obtained in the presence of ATP γ S, which has been widely used in the structural characterization of Drg1 related ATPases. In the manuscript the authors suggest that the slowly hydrolysable ATP analogue ATP γ S traps Drg1 conformations that would also occur in the presence of ATP. However, in some AAA+ proteins ATP γ S does not fully trigger the conformational changes normally observed in the presence of ATP. With respect to the SPR pre-60S release assays it is stated that ".....no significant release occurred in the presence of the slow-hydrolyzable analog ATP γ S....." (line 498, page 27). The authors should comment on this issue in the manuscript and ideally provide evidence that the use of ATP γ S does not affect Drg1 functionality in general, i.e. that Drg1 - even though the release rates might be much slower compared to ATP - is in principle still able to remove Rpl24 from pre60S particles in the presence of ATP γ S.

- Conformational rearrangements within the pre60S particle driven by Drg1 binding alone: In the manuscript the authors should clearly exclude in the possibility that the reported pre60S particle rearrangements upon Drg1 binding to pre60S particles are just "breathing effects". Were the Drg1 free pre60S particles in the data set analysed to distinguish intrinsic flexibility from rearrangements driven by Drg1 binding?

- Resolution of the Drg1-pre60S recruitment interface: In the complex, the density for Drg1 is blurry and flexible, which is a common problem for proteins at the periphery of multi-protein complexes. The authors provide local resolution maps in the supplement, but a qualitative measure for the resolution of the in the Drg1-pre60S particle interface should be mentioned in the main text (docked crystals structures or domains, resolved main-chain etc..)

- Drg1 hand-over-hand mechanism: Four AAA+ domains in the D1 and D2 rings seem to be engaged with the Rpl24 substrate. Other cryoEM structures of type II AAA+ unfoldases show five substrate engaged AAA+ domains at the (see Ripstein et al., "Structure of a AAA+ unfoldase in the process of unfolding substrate", eLife, 2017 or Cooney & Han et al., "Structure of the Cdc48 segregase in the act of unfolding an authentic substrate", Science, 2019). Maybe the authors could comment on these differences.

Author Rebuttal to Initial comments

Response to Reviewers' Comments:

Reviewer #1:

Remarks to the Author:

Prattes et al. (NSMB-A45578) report cryo-EM structures of an export-competent pre-60S particle, purified via the Bud20-TAP bait from cells treated with the export inhibitor leptomycin B, that either contain or lack exogenously added Drg1, a hexameric type II AAA-ATPase that was previously shown by the Bergler laboratory to mediate the release of the biogenesis factor Rlp24 from cytoplasmic pre-60S particles. The visualized structures of Drg1-bound pre-60S particles, together with the high-resolution structure of free Drg1 in the substrate-processing state, reveal how: 1) Drg1 is recruited, via the N-terminal domain of one of the six subunits, to its pre-60S docking site, which is mainly composed of Arx1 and rRNA expansion segment ES27; 2) the tips of four N-terminal domains contact the Rlp24 substrate and guide its C-terminal residues to the central pore of the hexameric ring-like structure; 3) Drg1 mechanistically exerts substrate threading through its central pore.

While the revealed substrate threading mechanism of Drg1 is similar to the one of the related AAA-ATPase Cdc48/p97, which is recruited to and segregates a broad range of ubiquitinated substrate proteins from membranes and stable complexes, it is novel with respect to how such an AAA-ATPase specifically associates with a macromolecular complex in order to be ideally positioned to extract its exclusive substrate protein. Notably, this study not only reveals how Drg1 achieves extraction of Rlp24 (and the neighboring Bud20) from cytoplasmic pre-60S subunits, but also indicates how this enforces larger conformational changes that prime the pre-60S subunit for downstream maturation events. Taken together, this study provides for the first time molecular insights into the mechanism of Drg1 action and highlights how this AAA-ATPase initiates the cascade of cytoplasmic pre-60S maturation events. These findings are not only exciting for the ribosome biogenesis field but are also of importance for our general understanding of the modus operandi of type II AAA-ATPases.

We thank reviewer 1 for the feedback and address all raised points in detail below!

Comments and suggestions:

1) Figure 2c: Based on the cryo-EM structure, one of the N-terminal domains of Drg1, by interacting with Arx1 and ES27, anchors the AAA-ATPase at the pre-60S particle in proximity of Rlp24. To investigate the importance of these interactions for recruitment of Drg1, it could be tested whether absence of Arx1 in combination with mutation of Drg1's ES27-interacting residues (K146, K150, K219) results in a synergistic growth defect.

We indeed observe a synergistic effect upon *arx1* deletion with mutations in the RNA-binding cleft of the Drg1 N-domain. We observe a mild growth phenotype when we mutate K219 together with R218 and a strong synergistic effect when we additionally mutate K117 and K118, suggesting that the loss of individual interacting residues can be tolerated, but combining the mutations strongly affects the interaction. No effect was observed for either K146 or K150 alone. Thus, a partially redundant set of residues seems to be responsible for the interaction. This is in line with our observation that Drg1

dynamically associates with the particle and thus the interaction surface is not completely static and involves a network of residues. We included the new result (Fig. S3d) in the revised manuscript.

2) Results, line 284 and legend to Figure 3c: Based on the pronounced dominant negative growth phenotype elicited by ectopic expression of a Drg1 protein lacking the first 28 residues, the authors conclude that the “N-terminally truncated Drg1 variants are nonfunctional” (line 284) and that “this part of the protein is essential” (legend Figure 3c, line 267/268). In order to unambiguously show that the tip region of the N-terminal domain indeed fulfils an essential role for substrate recognition and guiding the C-terminal residues of Rlp24 into the D1 ring, the ability of the Drg1 Δ N20 and Drg1 Δ N28 variants to complement the absence of Drg1 should be assessed (as done in Figure 3e for the C-terminally truncated Rpl24 proteins).

We performed the suggested experiment and found that the N-terminal truncation of Drg1 by 28 amino acids (Δ N28) is only partially functional and shows reduced growth, when expressed from its endogenous promotor on a centromeric plasmid in the Δ drg1 shuffle strain (Fig. 3d). This indicates that the N-domain tips are important for substrate processing (for which we also provide evidence with our structural and crosslinking data), albeit they are not essential. To provide more evidence for the importance of the N-tips, we furthermore included an ATPase activity assay (Fig. 3e) that shows that the N-terminal truncation of 20 amino acid residues of Drg1 (Δ N20) affects its stimulation by Rlp24, while the basal activity remains on the wildtype level (Fig. 3e). We could not obtain the Drg1 Δ N28 variant in sufficient quality and purity for this analysis. We included the new data in the manuscript and rephrased the section accordingly.

3) Results, line 288-290: The authors state that “Thus, the very C-terminal end of Rlp24 is crucial for interaction and stimulation of Drg1.” and then conclude that “Due to the specificity of Drg1 for Rlp24, the recruitment has to be sequence specific.”. First, it has been previously shown that Rlp24 Δ C, lacking the complete C-terminal domain, still interacts with Drg1 (Kappel et al. J. Cell Biol. 2012).

While we observed some interaction of the Rlp24 Δ C with Drg1 *in vitro* (Kappel et al., 2012), this fragment of Rlp24 does not stimulate the ATPase activity of the AAA-ATPase. The interaction of Rlp24 Δ C therefore has to be of different nature than that of the Rlp24 C-domain with Drg1. In support of our suggestion of sequence specific characteristics in the recognition between Drg1 and Rlp24C, we meanwhile observed that exchanging 3 basic residues in the tail of the Rlp24C-terminal domain is sufficient to reduce recognition by Drg1 and stimulation of its ATPase activity. This indicates that not only the length of the unstructured C-domain is important, but that sequence characteristics play a role in recognition and stimulation of Drg1 and Rlp24. Nevertheless, the residual activity seems to be sufficient to support growth. We included the respective new experiments in Fig. 3f-i. However, we agree with reviewers 1 and 2 that “has to be sequence specific” is a too strong wording. We therefore changed it to “shows sequence specific characteristics”.

Second, assuming that the human Drg1 ortholog SPATA5 would also mediate extraction of the Rlp24 ortholog in a similar manner, it is surprising that the C-terminal domain/tail is not conserved between the yeast and human Rlp24 protein.

There is still very little information about the exact function of SPATA5 in human ribosome biogenesis (Tafforeau *et al.*, 2013, Mol Cell). In the structure of the human pre-ribosome (e.g. PDB 6LSS, Liang *et al.*, 2020, Nat Commun), hRlp24 is only resolved up to residue 136 (of 163). We therefore expect that the C-terminal end of hRlp24 is protruding from the ribosome in a similar manner as in

yeast. Indeed, we observe *in vitro* binding between human Rlp24 and SPATA5 in GST pulldown assays suggesting conservation of function (**Fig. A**). Interestingly, SPATA5 has an about 100 amino acids insertion in the N-domain and an about 15 residues prolonged N-tip. Considering the important role of the N-terminal domain of Drg1 in Rlp24 recognition, we speculate that this N-terminal extension of SPATA5 could compensate for the altered characteristics of human Rlp24. However, due to the limited (experimental) information about these two proteins in humans, this remains merely speculative and is an interesting target for future work.

Fig. A: GST-hRlp24 pulldown with SPATA5.

Further, how do the authors reconcile the proposed sequence-specific recognition and threading of Rlp24's C-terminal tail with the observation that strains expressing C-terminally TAP-tagged Rlp24 have been reported to be viable (Saveanu et al. Mol. Cell. Biol. 2003, Pertschy et al. Mol. Cell. Biol. 2007). Can Drg1, like Cdc48 (ubiquitin), also process stably-folded substrates? To confirm that Rlp24-TAP is indeed (partially) functional, I encourage the authors to assess complementation of the Δ rlp24 shuffle strain by Rlp24-TAP expressed from plasmid. At the same time, it would also be informative to test C-terminally GFP-tagged Rlp24 for complementation.

We generated the suggested C-terminally TAP- and GFP-tagged Rlp24 versions and found that, in our strain background, centromeric plasmid borne Rlp24-TAP exhibits a strong growth phenotype, but is still partially functional (**Fig. B, left panel**). In contrast, Rlp24-GFP cannot complement the chromosomal deletion. To our knowledge, the only report for a chromosomally GFP-tagged Rlp24 was from the mentioned paper by Saveanu and coworkers in 2003. The respective strain, however, did not regrow after freezing (Micheline Fromont-Racine, personal communication). This also suggests that the GFP-fusion is actually nonfunctional. In contrast, the Rlp24-TAP genomic fusion was used by Micheline's Lab and our Lab and is (at least partially) functional.

Regarding the question how the C-terminally TAP-tagged version of Rlp24 can be processed by Drg1, we could envisage extraction by a similar mechanism as recently described for p97/VCP in Van den Boom et al., 2021 (NSMB) (**Fig. B, right panel**) where a hairpin peptide chain is pulled into the central pore of the human VCP protein that is closely related to Drg1. This shows that the central pore of VCP

(and possibly also Drg1) can accommodate two peptide chains, in this case in the form of a loop. Since the substrate translocation channels of Drg1 and VCP are highly similar (see new subpanel in supplemental Fig. S4e-f in our revised manuscript for comparison of Drg1 with the yeast VCP ortholog Cdc48), we speculate that a similar hairpin formation could enable processing of the TAP-tagged version of Rlp24. The better survival of the Rlp24-TAP strain at 37°C could thereby reflect a beneficial effect of higher temperature on unfolding of the Protein A-tag during extraction of Rlp24-TAP from the pre-ribosome. In contrast, the very stably folded GFP-tag presumably cannot be accommodated or unfolded in this way and therefore does not support growth.

Fig. B. Left panel: Expression of C-terminally tagged Rlp24 variants from centromeric plasmids under the control of their endogenous promoter in a $\Delta rlp24$ shuffle strain. **Right panel:** processing of a hairpin structure by p97 (taken from Van den Boom *et al.*, 2021 (NSMB))

4) Crosslinking MS (Supplementary Table S2): How do the authors explain that crosslinks between Drg1 and early biogenesis factors (Drs1, Erb1, Nop8, etc.), which are not expected to be present on the same pre-ribosomal particles as Drg1, are detected in these experiments? Does this question the specificity of the reported crosslinking data?

Crosslinking MS has been carried out according to an established protocol with a proven track-record for yielding high-confidence and specific crosslinks, in particular for ribosomal complexes (Erzberger *et al.*, 2014, Cell, PMID: 25171412; Obayashi *et al.*, 2017, Cell Rep., PMID: 28297669) and ribosomal intermediates (Kargas *et al.*, 2019, Elife, PMID: 31115337). However, even the most diligent list of cross-links will contain a certain number of false-positive identifications. Most crosslinks present in our experiments are in line with known interactions (e.g. Arx1-Alb1, L24-Rei1) or published and unpublished pre-ribosome structures. Finally, we observe the same Rlp24C crosslinks to Drg1 K24 and K13 when probed only with the Rlp24 C-domain fused to GST and also using different crosslinkers (Ruth Birner-Grünberger, Gertrude Zisser and Helmut Bergler, unpublished results). We are therefore convinced that minor contaminations with some early maturation factors do not compromise our results.

5) Figure 1a: Why is Drg1 present in good amounts on Bud20-TAP pre-60S particles isolated from leptomycin B-treated cells? Should the isolated particles not correspond to nuclear pre-60S particles whose export to the cytoplasm is inhibited and should therefore not contain Drg1? Can the authors provide an explanation for this observation?

The western blots were exposed to show similar densities in the untreated lanes of all blots to support comparison of enrichment and depletion of maturation factors in the different samples (we added a sentence to highlight this fact to the legend of Fig. 1). The strong signal for Drg1 therefore does not represent a large amount of the protein in the sample. The cellular level of Bud20 is approx. 4000 molecules while there are only about 130 hexamers (~800 molecules) of Drg1 in a yeast cell (Ghaemmaghami *et al.*, 2003 (Nature)) which means that only few of the isolated Bud20-TAP particles contain Drg1. After LmB treatment, the particles containing Drg1 are further reduced. The still detectable Drg1 levels likely arise from incomplete inhibition of Crm1 which is indicated by residual levels of Crm1 in the LmB treated sample in Fig. 1a and by the cytosolic staining of the Bud20-GFP fusion after simultaneous treatment with LmB and diazaborine (Fig 1b).

6) Discussion, line 540-542: The authors mention only briefly once that loosening of Rlp24 from the complex may involve an entropic pulling force generated by the movement of Drg1 relative to the particle. Can the authors provide references to articles here that highlight how ATPases utilize entropic pulling forces to act on substrates? On the other hand, the phrasing of many other sentences suggests that Drg1 uses mechanical force or acts similarly as a 'Brownian Ratchet' to extract Rlp24: "The coordinated consecutive grabbing and releasing of the substrate by the pore loops creates directionality and stepwise pulls the substrate chain through the pore. (line 447-449)", "The unphased behavior of the D1 and D2 rings likely guarantees a constant unidirectional force generation required to pull off Rlp24 from the pre-ribosome and thus counteract the peptide's chain backtracking. (line 574-576)" and "Taken together, ... and dissect the chain of events during mechanical extraction of its specific substrate. (line 602-604)". I feel that it might help to more clearly discuss the different possibilities, maybe they are not even mutually exclusive.

We fully agree with reviewer 1 that it would be helpful for the reader to provide more information on this topic. We now briefly address the different mechanistic models of substrate processing discussed in the community and provide suitable references.

7) Discussion, line 566-568: The authors mention only briefly that Drg1-mediated extraction of Rlp24 "... follows a conserved scheme in type II AAA-ATPases.". If possible, can they provide a Supplementary Figure comparing (or overlaying) the substrate-processing Drg1 and Cdc48 structures to visualize the similarities?

We added a subpanel to Figure S4 (e-f) showing a superposition of substrate-bound Drg1 and Cdc48 (PDB: 6OPC) as suggested and included a statement in the main text. The superposition demonstrates that the substrates inserted into the pores of the two proteins are almost perfectly superposable with the same corkscrew-like arrangement (even in the presence of different nucleotides) strengthening the hypothesis of a general translocation mechanism based on the conserved geometry of the pore. Interesting differences are however found in the overall conformation of the Cdc48 hexamer which shows a more pronounced staircase conformation. Furthermore, the N-domains of Drg1 are positioned closer to the bound substrate than in Cdc48. At this stage, however, we cannot distinguish if these differences display general mechanistic differences or reflect the different nucleotide binding state (ADP vs ATP γ S) or the presence of the adaptor Sph1 in the available Cdc48 structure.

Minor points:

- 1) Abstract, line 38 (and also later on): eukaryote-specific instead of 'eukaryotic specific'
- 2) Introduction, line 58: remove the definite article before 5.8S.
- 3) Introduction, line 65 (and also later on): Crm1/Xpo1 instead of 'Crm1/Xpo1'
- 4) Results, line 94-96: It might help to also state in the text (and not only in the legend to Figure 1a) that the visualized pre-60S particles were purified via Bud20-TAP. I would therefore suggest to rephrase the sentence as follows: "We assembled pre-60S particles, purified via the Bud20-TAP bait, with wild-type Drg1"
- 5) Results, line 145: Mrt4 instead of 'Mtr4'
- 6) Results, line 504-506: This sentence seems to be incomplete. It could be changed to: 'Taken together, our results show that Drg1 is highly efficient in releasing Rlp24 (and Bud20) from nascent pre-60S particles and that the release strictly depends on ATP hydrolysis in the D2 domain.'
- 7) Two of the used strains are not listed in the strain table: Bud20-GFP crm1.T539C (used in Figure 1b) and Arx1-TAP drg1-18 (used for crosslinking MS).

We thank reviewer 1 for drawing our attention to these points and corrected all of them!

Reviewer #2:

Remarks to the Author:

The manuscript "Visualizing maturation factor extraction from the nascent ribosome by the AAA-ATPase Drg1", by Prattes et al. provides a structural exploration into how Drg1 removes Rpl24 from pre-60S particles in order to drive ribosome maturation. Their data identify important domains and regions for substrate recognition and processing, and confirm the stereotypic AAA "hand-over-hand" model. Overall, the completed work is very impressive, and I believe it will be a useful addition to the field and is appropriate for publication in NSMB, after my concerns are addressed.

We thank reviewer 2 for the feedback and address all points in detail below!

Concerns/comments:

1. Figure 1a: Why is there signal for Drg1 in the Western Blot for LmB-treated cells? Are these gels run after overexpressed Drg1 was added? If not, the authors should discuss.

The particle samples on the gel do not contain purified Drg1, which was only added directly prior to cryo freezing. We rephrased the text to state this more clearly. The residual Drg1 levels in the LmB treated strain likely arise from incomplete inhibition of Crm1 which is obvious from residual levels of Crm1 in the LmB treated sample in Fig. 1a, as well as from cytosolic staining of Bud20-GFP after treatment of the strain with LmB and diazaborine simultaneously (Fig. 1b). As also outlined in the comments to reviewer 1, the western blots were exposed to show similar levels in the untreated lanes of all blots to facilitate comparison of the different samples (we now mention this in the Figure legend to prevent confusion). In fact, only few percent of particles from the untreated strain contain Drg1, which is further reduced in the LmB treated sample.

Further towards that end, the main text (section starting on line 92) should more explicitly say that you are adding separately-expressed and purified Drg1 to the native particles prior to freezing.

As suggested, we rephrased the respective sentence to make this clearer: “We assembled pre-60S particles, purified via Bud20-TAP as bait, with separately purified wild-type Drg1 *in vitro* and collected single-particle cryo-EM data.”

2. Line 175: This is too strong of an interpretation, and should either be moved to the discussion or removed. Most notably, the structure is in the presence of ATP γ S, not ATP (and reasonably so), where force is not being generated by the motor. The ring may very well be more “plush” against the pre-60S in ATP – we don’t know. Using only the ATP γ S structure, these claims are far too speculative for the Results section.

We agree with the reviewer that this belongs to the discussion. We moved it accordingly and also address possible influences of the nucleotide.

3. Fig 2D and the associated Results section (line 235) are opaque to me. It is difficult to fully assess if this variability analysis is sufficient to justify the strong claims on specific changes to protein-protein interactions. A more detailed explanation of the 3D variability analysis protocol used should be provided, and the limitations of this analysis should be discussed.

Based on the questions of the reviewer we now describe the 3D variability analysis in some more detail in the methods section. We also rephrased this part in the results section, discussed limitations of this method and also clearly stated in the discussion that the results provided by this approach have to be validated by future studies.

4. Line 282 and 334: The claim that four N-domains engage the substrate is too strong to be justified by the data. Engagements are only qualified to within the limits of the resolution of the structure. Only one nucleotide state – ATP γ S – is assessed. A different number of N-domains may engage in different ATP/ADP states. These limitations must be explicitly discussed, or the claim should be dropped.

We clearly see a structure that shows 4 N-domains in the close proximity of the substrate, however, at low resolution. Given a long unresolved N-terminal stretch before the folded part of the domain, and the seen space limitations, engagement of these domains with the substrate is very likely. The two other N-domains are in our structure found in a different orientation and different angle relative to the substrate. We agree, that this is a highly dynamic process and the number of interacting domains may of course be linked to the nucleotide state. We added a statement to underline that our observations are based on ATP γ S data and that different nucleotides could lead to other arrangements as suggested by the reviewer.

5. I am concerned that the claim “The Drg1 N-domain tips coordinate initial substrate recognition” is not fully justified by the data (Fig. 3 title, line 280, line 294, Fig. 5a). How do you know that the N-domain forms contacts prior the D1D2 domains temporally? There is no data to support this. Engagement is only showed in a single nucleotide state. It is perfectly possible that the N-domains don’t engage with the substrate until after it is already in the pore, and serve some important role (e.g., preventing slippage).

The finding that the N-domain tip (residues K13 and K24) crosslinks to the very C-terminal region of Rlp24 (K186) supports our hypothesis that the N-domains are (also) involved in the initial stage of substrate recognition before it is inserted in the pore. This contact would be unlikely when the C-domain is already fully inserted in the pore. However, we agree that we currently cannot

definitely resolve the stage at which the N-domain tips engage with the substrate temporally and thus rephrased the relevant part in the results as well as the discussion section accordingly also including other possibilities.

6. Figure 5 and Results Section “Substrate extraction via hand-over-hand translocation: It is not clear to this reviewer how information about nucleotide-dependent structural changes can be extracted from a 3D variability analysis of a single (ATP γ S) structure.

In our dataset we observed that the conformational changes of the hexamer correlate with the density of the nucleotide in the binding pocket indicative for association/dissociation of the nucleotide which affects the conformation of the hexamer. Furthermore, our data reflect global conformational changes and movements that correlate well with the coordinated association and dissociation of the pore loops with the substrate in course of the translocation process also documented for related AAA-ATPases (e.g. recently reviewed in Puchades et al., 2020). In addition, we find that the substrate is almost identically inserted into the pore as observed for Cdc48 again indicating that substrate translocation by Drg1 follows a conserved mechanism. We now included a comparison between Cdc48 and Drg1 in supplemental Fig. S4e-f to highlight this. These facts indicate that the observed dynamic conformational changes reflect the physiological range of motions that the Drg1 hexamer can go through depending on the nucleotide loading state. Indeed, we cannot fully assess the impact of the physiological level of nucleotide hydrolysis in the ATP bound state, but our data reflect the transitions between nucleotide bound and unbound (apo) state as we see the density for the nucleotide vanishing and the resulting structural changes (e.g. Movie 1). Although with ATP γ S the hydrolysis is very slow, association and dissociation will happen similarly as with ATP. In several studies mixtures of ATP γ S with ATP are used to slow down AAA-proteins suggesting that ATP γ S is not fully blocking but just slowing down (e.g. Doyle et al., 2007, (PNAS), Aubin-Tam et al., 2011 (Cell)). Additionally, we are convinced that the Brownian ratchet mode of action of molecular motors is the most likely model which is reflected in our manuscript, although we cannot definitely exclude other models. We now adapted the text at the respective positions to emphasize that our conclusions are derived from ATP γ S and nucleotide binding/dissociation data and discuss our data in respect to the different models.

7. Fig. 6C: The Rlp24C binding data does not plateau. In such a case, it is improper to normalize to the higher data point. A fitted EC50 and Hill sloped cannot be determined. All that can be determined is a lower bound for the EC50, i.e., the EC50 is $> 1 \mu\text{M}$. The Bud20 data barely plateaus. The authors should repeat this experiment at high Drg1 concentrations to ensure a stable upper plateau of binding.

In our original figure, we tried to depict the different binding strength of Drg1 to isolated Rlp24C and to the pre-60S particle in the same graph which however did not optimally represent the binding characteristics. Plotted in linear form, the Rlp24C binding curve does approach saturation. For better representation, we therefore now show the binding to Rlp24C, which follows a hyperbolic binding mode (i.e., Hill coefficient of 1), in a separate panel as the binding to the particle (which clearly shows a positive cooperative binding). In addition, as suggested by Reviewer 2, we measured concentrations of up to $10 \mu\text{M}$ of Drg1 for binding to Rlp24C and included these data in the revised manuscript.

8. Fig 6D: The release rate estimate depends strongly on the rough estimate of 1000 RU equaling 3 mg/mL. I am concerned that this estimate is inaccurate. Deposition of the particles leads to a response of 1,500-1,700 RU, which should accordingly be $\sim 4.5 \text{ mg/ML}$. What is the particle concentration in mg/mL? Is it reasonable to be injecting/depositing 4.5 mg/mL? Ideally, the authors will either devise a more quantitative way of converting RU to nM, or explicitly state the limitations and assumptions built

into their measurements, and appropriately pare back language to match the uncertainty. Furthermore, if deposition leads to 1,500 RU, why does addition of 75 nM Drg1+ATP lead to only -400 RU at the plateau? Presumably particles cannot rebind after being processed by Drg1.

The conversion of RU to the surface concentration that we used is based on an empirical determination using radiolabeled proteins of different size by the manufacturer (Pharmacia, Stenberg et al., 1991, Journal of Colloid and Interface Science)). This analysis showed that the specific response of a protein varies only very little in regard to the size of the protein. The corresponding formula for the calculation of the surface concentration (pmol/mm²) and the corresponding volume concentration (mol/l) which also includes the thickness of the matrix on the chip (Müller et al., 1998), is thus not just based on theoretical considerations. Specifically, the conversion from RU units to changes in analyte concentration depend in a linear fashion on two quantities, that is, the refractive index of the analyte and the mentioned thickness of the matrix on the SPR chip. Importantly, the refractive indices of biomacromolecules fall within a narrow range of values (Zhao et al., 2011, Biophys. J.). Similarly, the uncertainty in the physical thickness of the matrix are also supposed to be small.

Since particles lose the TAP-tagged Bud20 that was used to capture them on the SPR chip, they indeed cannot rebind to the chip once they are processed by Drg1 as suggested by reviewer 2. The finding that only 400 RU are released might indicate that the Drg1 binding site on a subset of particles captured on the Chip is not fully accessible for Drg1. We are aware of the complexity of the system and its limitations and aimed to bring this across more clearly now in the revised manuscript. Based on the query of the reviewer, we now described the basis for our conversion in the manuscript in more detail and referenced the underlying formula and background information. To further improve the quantification, we have now also adapted the calculation to better integrate the specific characteristics of the chip that we used in our setup. We updated the respective paragraphs in the results and methods section and furthermore moved the paragraph regarding the conversion to particle number and the relation to the number of molecules in the cell into the discussion where limitations can be addressed more adequately. The bottom line is that our approximate quantification of the release rate is consistent with the rate required for the cellular Drg1 pool to cope with its physiological function and reveal a reasonable agreement between *in vitro* and *in vivo* results.

Minor Comments

1. Line 61: Missing parenthesis

We added the parenthesis.

2. Line 92: Please be more explicit in stating the model organism, strain, and pre-60S isolation strategy in the main text.

We reformulated the text to add the needed information.

3. Line 137: The authors should discuss here the average resolution of the CryoEM structure. Is it sufficient to build in side chains, etc.

As usual in cryo-EM structures the resolution is not homogeneous in all parts of the map. This is due to the flexibility of individual components and domains. We obtained near-atomic resolution for the core of the pre-ribosomal particle in the complex and 3.2 Å for Drg1 alone, which allows good placement of side chains in most regions. However, in the complex, Drg1 smears outside of the attachment site to the pre-ribosomal particle. But due to the availability of models for all components, obtained previously or in this study, we can build a hybrid model. With this we are confident, that the backbone is placed correctly. However only a few side chains show density in the interface region. We

decided to leave them in the model, as it is also commonly done in many other studies. To our knowledge there is still no agreement in the structural community whether it is best to only build atoms that are seen in the density and thus produce fragmented models or produce complete models that however have some regions of lower confidence. We did the latter as this is more useful to the reader.

4. Figure 1a-b legend: Please define the “0” condition (first lane in gel).

We changed the lane description from “0” to “untreated” and adapted the figure legend accordingly.

5. Line 167: Please provide a quantitation of the “minor population” (e.g., relative # of particles). Separately, it would be helpful if the authors could add color to the important elements in the “additional contact site” box, similar to how they did for the “docking platform” box

We added the quantitation for the minor population. Furthermore, we adapted the figure as suggested.

6. Fig 2a: A linear domain diagram of Drg1 would be helpful, including aa cutoffs for the Nn and Nc subdomains.

We added the suggested domain diagram including cutoffs in Fig S3a.

7. Dig S3C: Please state in the caption what the data points represent (technical or biological replicates), as well as what the error bars represent (presumably mean \pm SD)

Thank you for drawing our attention to this missing information which is now included.

8. Line 202-203: What data support this claim? This statement should be in the Discussion not Results, with an appropriate qualifier

Due to the multi-domain structure of Drg1 we expect allosteric effects and it is also documented for Drg1 (e.g. Loibl *et al.*, 2014, JBC; Prattes *et al.*, 2017, Sci Rep) and for example p97 (e.g. Banerjee *et al.*, 2016, Science, Schütz and Kay, 2016, elife) that there is a mutual influence of the N-domains and the AAA-ATPase domains. We do also see correlated movements between the N-domain and the D-domains in our 3D variability analysis of Drg1. However, since we did not further address this specific question experimentally in this study we removed this claim.

9. Line 214: Please perform mutagenesis analysis to specifically and definitely prove that RNA contacts to the N-domain are essential, or remove this claim of primacy.

We analyzed mutations in the RNA binding cleft of the Drg1 N-domain as requested and find strong synergistic growth defects upon combining them with the *arx1* deletion. We included the analysis in Figure S3e and mention the result in the text.

10. Fig 3a caption: Please state the threshold used

We added the threshold.

11. Line 290: “has to be sequence specific” is uncomfortably strong language. Rlp24 is currently the known substate of Drg1, but others may exist and be uncharacterized.

To further address this question, we performed additional experiments showing that point mutations in the Rlp24 C-terminal tail (RKK>EEE) strongly affect the stimulation of the Drg1 ATPase activity and binding to Drg1. In contrast to the deletion of the C-terminal tail, this mutant variant of Rlp24 is still partially functional (likely since it is still able to stimulate the activity of Drg1, although to a much lesser extent). We included these experiments in Fig. 3g-i. These results corroborate our hypothesis that a stimulating substrate likely needs to provide some sequence specific characteristics and not only a specific length of a freely accessible unstructured polypeptide chain. However, we agree with the reviewer that this statement was not optimally phrased and thus changed the wording. We now rephrased the respective paragraphs to state that the recognition/stimulation “involves sequence specific features”.

12. Line 361: Again, please acknowledge in the text that this interpretation is only made based on ATP_γS data. Further work is needed to know if the D1 and D2 domains are significantly out of phase in the

We rephrased the sentence to make it clear that we describe the ATP_γS-bound state.

Reviewer #3:

Remarks to the Author:

The manuscript entitled “Visualizing maturation factor extraction from the nascent ribosome by the AAA-ATPase Drg1” reports a series of cryoEM structures of the maturing large ribosomal subunit in complex with the ribosome assembly factor Drg1. Drg1 is a type II AAA+ ATPase and catalyses the first pre60S particle maturation step after nuclear export, the removal of Rpl24. The authors use a chemical inhibitor to enrich Drg1 depleted pre60S particles and reconstitute the complex by adding recombinantly purified Drg1. The functionality of this in-vitro reconstituted complex - the ability to extract Rpl24 in the presence of ATP - is demonstrated by SPR release assays.

The authors visualize pre60S remodelling events caused by the binding of the Drg1 hexamer and are able to analyse the Drg1 recruitment interface formed by Arx1, ES27 rRNA and helix 98 of the 25S-RNA. The presented cryoEM structures of the complex also reveal the initial steps of Rpl24 capture by the Drg1 N-terminal domain. Using high-resolution cryoEM structures of free, substrate engaged Drg1 hexamers, they can analyse the mechanism of Rpl24 unfolding by the Drg1 hexamer. Similar to other AAA+ unfoldases, Rpl24 unfolding proceeds via a classical “hand-over-hand” mechanism, which involves loop regions harbouring conserved tyrosine residues that thread Rpl24 through the central pores of the AAA+ rings. The authors are also able to indirectly quantify the Drg1 mediated Rpl24 release rate from pre60S particles using a SPR approach.

All experiments seem solid and the presented conclusions are justified. The cryoEM maps have been carefully analysed and not over interpreted. The structural hypotheses developed based on the presented cryoEM maps are nicely supported by additional biochemical experiments: The Drg1 - pre60S recruitment interface is validated by crosslinking mass spectrometry, the nucleotide independency of the Arx1 – Drg1 interaction by pull down assays and the importance of the interaction between the C-terminus of Rpl24 and the N-terminal Drg1 domain for initial engagement by yeast growth assays and ATPase assays.

The manuscript highlights a key step in the maturation of the large ribosomal subunit at unprecedented detail for the first time. The pre60S-Drg1 and Drg1 structures provide novel insights into Drg1 recruitment to pre60S particles and the capture of its Rpl24 substrate. The presented results are an important contribution to ribosome maturation field and will be of great interest for the AAA+ community as well. I recommend publication in NSMB.

We thank reviewer 3 for the feedback! Please find a point-by-point response below.

Minor points:

- The presented structures were obtained in the presence of ATP γ S, which has been widely used in the structural characterization of Drg1 related ATPases. In the manuscript the authors suggest that the slowly hydrolysable ATP analogue ATP γ S traps Drg1 conformations that would also occur in the presence of ATP. However, in some AAA+ proteins ATP γ S does not fully trigger the conformational changes normally observed in the presence of ATP. With respect to the SPR pre-60S release assays it is stated that “.....no significant release occurred in the presence of the slow-hydrolyzable analog ATP γ S.....” (line 498, page 27). The authors should comment on this issue in the manuscript and ideally provide evidence that the use of ATP γ S does not affect Drg1 functionality in general, i.e. that Drg1 – even though the release rates might be much slower compared to ATP - is in principle still able to remove Rpl24 from pre60S particles in the presence of ATP γ S.

We rephrased several sentences in the main text to make it clearer that we describe the findings from the ATP γ S bound state. It is not expectable that this state fully reflects the ATP bound state, but to capture Drg1 mid-processing on the particle (in the cryo-EM as well as the SPR measurements) it was indispensable to use this slow-hydrolyzing nucleotide which is an established strategy to analyze substrate bound states of AAA-ATPases (e.g. Ripstein et al., 2017, Deville et al.,

2019, reviewed in Gates and Martin, 2019). Especially ATP γ S has been seen to allow more conformations in other studies (e.g. Gates et al. 2017, Science) Nevertheless, in our 3D variability analysis we detect characteristic conformational changes and coordinated movements of the individual protomers relative to each other as well as of the pore loops relative to the substrate that are in line with the proposed canonical substrate translocation model for type II AAA-ATPases (recently reviewed in Puchades et al., 2019). Thus, we conclude that ATP γ S allows Drg1 to perform its conformational changes needed for substrate recognition and at least partial translocation until the substrate is inserted into the pore, but due to the missing (or slowed down) hydrolysis of the nucleotide, the translocation process cannot be completed or is significantly slowed down compared to ATP. Thus, the basic activities and range of motions of the protein have to be preserved in the presence of ATP γ S. This also reflects the situation in the SPR measurements where Drg1 is able to bind to the particle as well as the Rlp24 C-domain alone, but significant release in the investigated timeframe only occurs in the presence of ATP.

- Conformational rearrangements within the pre60S particle driven by Drg1 binding alone: In the manuscript the authors should clearly exclude in the possibility that the reported pre60S particle rearrangements upon Drg1 binding to pre60S particles are just “breathing effects”. Were the Drg1 free pre60S particles in the data set analysed to distinguish intrinsic flexibility from rearrangements driven by Drg1 binding?

ES27 was, due to its mobility, not detected in a preliminary additional dataset recorded without Drg1. This corroborates that ES27 is stabilized in the presence of Drg1 as shown in movie S1. Our data clearly show a rotation of ES27 upon Drg1 binding, which is hard to explain by breathing effects that should occur more randomly and globally. Moreover, since ES27 is as part of the 25S rRNA covalently linked to the main body of the pre-60S particle its rotation is likely to affect the structure of pre-ribosome. Indeed, ES27 is proposed to provide a physical link between different functional sites of the ribosome (we included the relevant literature into the discussion of the revised manuscript). The detailed analysis of all changes seen in our 3D variability analysis is out of the scope of this manuscript. We therefore rephrased the text at several positions to highlight this for the reader. Nevertheless, these results can provide novel perspectives for the investigation of early steps in the cytoplasmic pre-60S maturation and the engagement of AAA-ATPases with large, macromolecular targets. To provide more information for the reader, we now also discuss limitations of the 3D variability analysis and clearly state that these results only provide a starting point for more detailed analyses in the future.

- Resolution of the Drg1-pre60S recruitment interface: In the complex, the density for Drg1 is blurry and flexible, which is a common problem for proteins at the periphery of multi-protein complexes. The authors provide local resolution maps in the supplement, but a qualitative measure for the resolution of the in the Drg1-pre60S particle interface should be mentioned in the main text (docked crystals structures or domains, resolved main-chain etc..)

Inspired by the questions of the reviewer, we have improved our local resolution map including an additional cross section to better visualize the distribution of the resolution over the whole particle. The interface indeed is blurry in the overall structure and we used several masked and classified maps to model this region as accurate as possible. To achieve a consistent model, in a first step rigid body fits of models were added that were manually adjusted wherever possible. Only a few interacting amino acids are resolved in the map to a full extend. We however placed the side chains coming from our own structure (Drg1 alone) or previously solved structures (ARX1, ES27). The quality of the map in this region is unambiguous to place the backbone accurately, thus the placement error of the side

chain atoms is supposed to be small. Therefore, we decided to keep these side chains to provide a complete model.

- Drg1 hand-over-hand mechanism: Four AAA+ domains in the D1 and D2 rings seem to be engaged with the Rpl24 substrate. Other cryoEM structures of type II AAA+ unfoldases show five substrate engaged AAA+ domains at the (see Ripstein et al., "Structure of a AAA+ unfoldase in the process of unfolding substrate", eLife, 2017 or Cooney & Han et al., "Structure of the Cdc48 segregase in the act of unfolding an authentic substrate", Science, 2019). Maybe the authors could comment on these differences.

In our structure, four AAA-domains are engaged in substrate binding in the D1 domain, while five are engaged in the D2 domain of Drg1. In any structural analysis of a dynamic molecular process, we will only capture snapshots of all possible interactions. Still, it is indeed an interesting finding that we presumably captured a transition state where the two AAA-domains of one protomer are asynchronously associated with the substrate. This might represent an intermediate state during the displacement of the protomers while grabbing along the substrate. We added a paragraph addressing this topic and its relationship to other structures in the discussion section. Further on this topic, we have now also included a superposition of Drg1 with Cdc48 to display similarities and differences between these proteins in the substrate-bound state (supplemental Fig. S4e-f)

This email has been sent through the Springer Nature Tracking System NY-610A-NPG&MTS

Decision Letter, first revision:

Our ref: NSMB-A45578A

29th Mar 2022

Dear Dr. Bergler,

Thank you for submitting your revised manuscript "Visualizing maturation factor extraction from the nascent ribosome by the AAA-ATPase Drg1" (NSMB-A45578A). It has now been seen by the original referees and their comments are below. The reviewers find that the paper has improved in revision, and therefore we'll be happy in principle to publish it in Nature Structural & Molecular Biology, pending minor revisions to satisfy the referees' final requests and to comply with our editorial and formatting guidelines.

We are now performing detailed checks on your paper and will send you a checklist detailing our editorial and formatting requirements in about a week. Please do not upload the final materials and make any revisions until you receive this additional information from us. However, you could use this time to already start preparing the revisions in response to the minor points remaining from Reviewer #1.

To facilitate our work at this stage, we would appreciate if you could send us the main text as a word file. Please make sure to copy the NSMB account (cc'ed above).

Thank you again for your interest in Nature Structural & Molecular Biology. Please do not hesitate to contact me if you have any questions.

Sincerely,
Sara

Sara Osman, Ph.D.
Associate Editor
Nature Structural & Molecular Biology

Reviewer #1 (Remarks to the Author):

The authors have satisfactorily addressed my comments and suggestions. I therefore support publication of the revised manuscript (NSMB-A45578A) in Nature Structural & Molecular Biology. However, the following two points should still be considered:

1) Figure 2b and 2c: The authors state (page 11, line 217-219) that "The ES27 loop is in close proximity to multiple lysine residues (117, 118, 146, 150 and 219) positioned in a groove in the N-domain (Fig. 2b and c)."; however, the position of residues K117 and K118, which have been mutated to glutamate to assess synergistic growth defects in the absence of Arx1 (Figure S3e), is not indicated

in Figure 2c. Therefore, these residues should be highlighted in Figure 2c. Moreover, the location of residues K146 and K150 within the N-domain groove should be indicated in Figure 2b. Further, a longer incubation of the 5-FOA plate at 25°C should be shown in Figure S3e (ARX1/DRG1 colonies should have the same size as on the 5-FOA 30°C plate) as the potential growth differences between *arx1Δ* single mutant and *arx1Δ drg1* double mutant cells (especially the combination with *drg1-R218E/K219E* and *drg1-K117E/K118E*) are barely visible at this temperature.

2) Figure 3d: To perceive the growth defects of the *drg1ΔN20* and *drg1ΔN28* mutants, a longer incubation of the plates must be shown (colony size of DRG1 complemented cells should be the same as the one of the wild-type control cells in Figures 3c and 3h). To better assess the impact of these N-terminal deletions, the growth on YPD plates at 23, 30 and 37°C (after 5-FOA) should be documented, this could be shown in a Supplementary Figure. Additionally, it would be informative to assess and compare the expression levels of Drg1 and the two N-terminal deletion variants.

Reviewer #2 (Remarks to the Author):

The authors have addressed my concerns and I recommend the manuscript for publication.

Reviewer #3 (Remarks to the Author):

All points raised in my initial review have been addressed. I recommend publication.

Decision Letter, final checks:

Our ref: NSMB-A45578A

17th May 2022

Dear Dr. Bergler,

Thank you for your patience as we've prepared the guidelines for final submission of your Nature Structural & Molecular Biology manuscript, "Visualizing maturation factor extraction from the nascent ribosome by the AAA-ATPase Drg1" (NSMB-A45578A). Please carefully follow the step-by-step instructions provided in the attached file, and add a response in each row of the table to indicate the changes that you have made. Ensuring that each point is addressed will help to ensure that your revised manuscript can be swiftly handed over to our production team.

If you have not done so already, please alert us to any related manuscripts from your group that are

under consideration or in press at other journals, or are being written up for submission to other journals (see: <https://www.nature.com/nature-research/editorial-policies/plagiarism#policy-on-duplicate-publication> for details).

In recognition of the time and expertise our reviewers provide to Nature Structural & Molecular Biology's editorial process, we would like to formally acknowledge their contribution to the external peer review of your manuscript entitled "Visualizing maturation factor extraction from the nascent ribosome by the AAA-ATPase Drg1". For those reviewers who give their assent, we will be publishing their names alongside the published article.

Nature Structural & Molecular Biology offers a Transparent Peer Review option for new original research manuscripts submitted after December 1st, 2019. As part of this initiative, we encourage our authors to support increased transparency into the peer review process by agreeing to have the reviewer comments, author rebuttal letters, and editorial decision letters published as a Supplementary item. When you submit your final files please clearly state in your cover letter whether or not you would like to participate in this initiative. Please note that failure to state your preference will result in delays in accepting your manuscript for publication.

Cover suggestions

As you prepare your final files we encourage you to consider whether you have any images or illustrations that may be appropriate for use on the cover of Nature Structural & Molecular Biology.

Nature Structural & Molecular Biology has now transitioned to a unified Rights Collection system which will allow our Author Services team to quickly and easily collect the rights and permissions required to publish your work. Approximately 10 days after your paper is formally accepted, you will receive an email in providing you with a link to complete the grant of rights. If your paper is eligible for Open Access, our Author Services team will also be in touch regarding any additional information that may be required to arrange payment for your article.

Please note that *Nature Structural & Molecular Biology* is a Transformative Journal (TJ). Authors may publish their research with us through the traditional subscription access route or make their paper immediately open access through payment of an article-processing charge (APC). Authors will not be required to make a final decision about access to their article until it has been accepted. [Find out more about Transformative Journals](https://www.springernature.com/gp/open-research/transformative-journals)

Authors may need to take specific actions to achieve  > **compliance with funder and institutional open access mandates**. If your research is supported by a funder that requires immediate open access (e.g. according to [Plan S principles](https://www.springernature.com/gp/open-research/plan-s-compliance)) then you should select the gold OA route, and we will direct you to the compliant route where possible. For authors selecting the subscription publication route, the journal's standard licensing terms will need to be accepted, including [self-archiving policies](https://www.springernature.com/gp/open-research/policies/journal-policies). Those licensing terms will supersede any other terms that the author or any third party may assert apply to any version of the manuscript.

Please use the following link for uploading these materials:
[Redacted]

Best regards,

Sophia Frank
Editorial Assistant
Nature Structural & Molecular Biology
nsmb@us.nature.com

On behalf of

Sara Osman, Ph.D.
Associate Editor
Nature Structural & Molecular Biology

Reviewer #1:

Remarks to the Author:

The authors have satisfactorily addressed my comments and suggestions. I therefore support publication of the revised manuscript (NSMB-A45578A) in Nature Structural & Molecular Biology. However, the following two points should still be considered:

1) Figure 2b and 2c: The authors state (page 11, line 217-219) that "The ES27 loop is in close proximity to multiple lysine residues (117, 118, 146, 150 and 219) positioned in a groove in the N-domain (Fig. 2b and c)."; however, the position of residues K117 and K118, which have been mutated to glutamate to assess synergistic growth defects in the absence of Arx1 (Figure S3e), is not indicated in Figure 2c. Therefore, these residues should be highlighted in Figure 2c. Moreover, the location of

residues K146 and K150 within the N-domain groove should be indicated in Figure 2b. Further, a longer incubation of the 5-FOA plate at 25°C should be shown in Figure S3e (ARX1/DRG1 colonies should have the same size as on the 5-FOA 30°C plate) as the potential growth differences between *arx1Δ* single mutant and *arx1Δ drg1* double mutant cells (especially the combination with *drg1-R218E/K219E* and *drg1-K117E/K118E*) are barely visible at this temperature.

2) Figure 3d: To perceive the growth defects of the *drg1ΔN20* and *drg1ΔN28* mutants, a longer incubation of the plates must be shown (colony size of DRG1 complemented cells should be the same as the one of the wild-type control cells in Figures 3c and 3h). To better assess the impact of these N-terminal deletions, the growth on YPD plates at 23, 30 and 37°C (after 5-FOA) should be documented, this could be shown in a Supplementary Figure. Additionally, it would be informative to assess and compare the expression levels of Drg1 and the two N-terminal deletion variants.

Reviewer #2:

Remarks to the Author:

The authors have addressed my concerns and I recommend the manuscript for publication.

Reviewer #3:

Remarks to the Author:

All points raised in my initial review have been addressed. I recommend publication.

Final Decision Letter:

3rd Aug 2022

Dear Dr. Bergler,

We are now happy to accept your revised paper "Visualizing maturation factor extraction from the nascent ribosome by the AAA-ATPase Drg1" for publication as an Article in Nature Structural & Molecular Biology.

As soon as your article is published, you can generate your shareable link by entering the DOI of your article here: http://authors.springernature.com/share. Corresponding authors will also receive an automated email with the shareable link

Your paper will be published online soon after we receive proof corrections and will appear in print in the next available issue. You can find out your date of online publication by contacting the production team shortly after sending your proof corrections. Content is published online weekly on Mondays and Thursdays, and the embargo is set at 16:00 London time (GMT)/11:00 am US Eastern time (EST) on the day of publication. Now is the time to inform your Public Relations or Press Office about your paper, as they might be interested in promoting its publication. This will allow them time to prepare an accurate and satisfactory press release. Include your manuscript tracking number (NSMB-A45578B) and our journal name, which they will need when they contact our press office.

About one week before your paper is published online, we shall be distributing a press release to news organizations worldwide, which may very well include details of your work. We are happy for your institution or funding agency to prepare its own press release, but it must mention the embargo date and Nature Structural & Molecular Biology. If you or your Press Office have any enquiries in the meantime, please contact press@nature.com.

Please note that *Nature Structural & Molecular Biology* is a Transformative Journal (TJ). Authors may publish their research with us through the traditional subscription access route or make their paper immediately open access through payment of an article-processing charge (APC). Authors will not be required to make a final decision about access to their article until it has been accepted. Find out more about Transformative Journals

Authors may need to take specific actions to achieve compliance with funder and institutional open access mandates. If your research is supported by a funder that requires immediate open access (e.g. according to Plan S principles) then you should select the gold OA route, and we will direct you to the compliant route where possible. For authors selecting the subscription publication route, the journal's standard licensing terms will need to be accepted, including self-archiving policies. Those licensing terms will supersede any other terms that the author or any third party may assert apply to any version of the manuscript.

Sincerely,
Sara

Sara Osman, Ph.D.
Associate Editor
Nature Structural & Molecular Biology
